# Global Gauge Symmetries and Spatial Asymptotic Boundary Conditions in Yang-Mills Theory

**Silvester G.A. Borsboom**[1][*] **and Hessel B. Posthuma**[2][†]

**1** Institute for Mathematics, Astrophysics and Particle Physics, Radboud University
**2** Korteweg-de Vries Institute for Mathematics, University of Amsterdam

[*] silvester.borsboom@ru.nl ,   [†] h.b.posthuma@uva.nl

## Abstract

In Yang-Mills theory on a Euclidean Cauchy surface, the physical gauge group is often taken to be $\mathcal{G}^I/\mathcal{G}_0^\infty$, where $\mathcal{G}^I$ consists of boundary-preserving gauge transformations asymptoting to a constant, and $\mathcal{G}_0^\infty$ of transformations generated by the Gauss law constraint. We rigorously derive this physical gauge group for both Abelian and non-Abelian theories. A key result is that restricting to $\mathcal{G}^I$ follows not just from finite energy, but from requiring the Yang-Mills Lagrangian be defined on the tangent bundle to configuration space. We extend our analysis to Yang-Mills-Higgs theory, showing that boundary conditions and the physical gauge group differ between the unbroken and broken phases.[1]

## Contents

---

[1]This article grew out of the master thesis of the corresponding author, supervised by the other author and by Sebastian de Haro [1].

## 1 Introduction

The physical status of gauge symmetries is a central topic in contemporary physics, both in Yang-Mills theory and general relativity. The term "gauge" is sometimes used as a synonym for "unphysical" or "empirically insignificant," but gauge transformations can acquire a physical meaning in the presence of boundaries [2–10]. A well-known empirical example is the Josephson current flowing between two superconductors whose boundaries are brought close together [11]. This current depends only on the relative difference between the *global U(1)* phases of the superconductors' Ginzburg-Landau order parameters, suggesting that global gauge symmetries are physical. Similarly, some gauge symmetries are physical on *asymptotic* boundaries. For instance, the asymptotic symmetry group of gravity in asymptotically flat spacetimes is the well-known BMS group [12–14] and asymptotic symmetries of Yang-Mills fields on both the null and spatial conformal boundaries of Minkowski spacetime are studied in the context of celestial holography, see e.g. [10, 15–19]. The general idea is that the asymptotic symmetry group consists of all "allowed symmetries" quotiented by all "trivial symmetries" [16]. Here "allowed" means those symmetries that respect the boundary conditions of the system and "trivial" means those symmetries that have no physical effect on the system. In this article we will identify the trivial symmetries of a gauge theory as those transformations that are generated by the Hamiltonian constraints of the theory.

Our aim is to rigorously derive the quotient of boundary-preserving gauge symmetries by trivial gauge symmetries for the specific case of Yang-Mills and Yang-Mills-Higgs theory on a Cauchy surface $\Sigma$ isomorphic to $\mathbb{R}^3$. Our motivation to do so comes from the desire to understand the physical content of the Higgs mechanism [20–23], which is thought to happen at a particular instant in time during the electroweak phase transition. For this reason we introduce a 3+1 split $\Sigma \times \mathbb{R}$ of spacetime and discuss *instantaneous* spatial asymptotic symmetries, for which the time $t$ is held fixed and the radial coordinate $r$ on $\Sigma$ is taken to infinity. This means that we do not consider the asymptotic symmetry group of full spatial infinity of Minkowski spacetime,[2] but only one "instant." It is sometimes said that asymptotic analyses are more of an art than a science [16, p. 34], but for the specific case of Yang-Mills theory on a Euclidean Cauchy surface we will present a fairly algorithmic and unambiguous method for deriving the physical gauge group from the assumption of finite energy. It is probable that this method can be extended at least to Yang-Mills theory on Cauchy surfaces in other spacetimes than Minkowski, and perhaps also to the gravitational field itself.

The case of Maxwell theory, possibly with a Higgs field, on Euclidean space has been studied extensively in the foundations of physics community, see e.g. [6, 7, 24–33]. The terminology used there to describe physical and trivial gauge symmetries respectively is that of *direct empirical significance* (DES) and *redundant* gauge transformations [28]. We will stick to this terminology. Redundant gauge transformations are contrasted with *formal* gauge transformations, which make up the full infinite-dimensional gauge group $\mathcal{G}$ without any regard for their physical status. For pure electromagnetism on $\Sigma$ with spatial asymptotic boundary conditions, the group of gauge symmetries carrying DES has been identified as the asymptotic symmetry group

$$\mathcal{G}_{\mathrm{DES}} = \mathcal{G}^I / \mathcal{G}_0^\infty,$$

---

[2]Spatial infinity understood as the timelike boundary at which spacelike geodesics end connects the infinite past with the infinite future, and is therefore itself infinitely extended in time and not instantaneous.

59 where $\mathcal{G}^I$ denotes the subgroup of the formal gauge group $\mathcal{G}$ whose elements leave asymp-
60 totic boundary conditions invariant[3] (the "allowed" symmetries), and $\mathcal{G}_0^\infty$ is the subgroup of
61 redundant gauge transformations that are generated by the primary first-class constraints of
62 the theory (the "trivial" symmetries). Here the $\infty$-superscript stands for the trivial action
63 of these transformations at infinity[4] and the subscript 0 denotes the identity component of
64 $\mathcal{G}^\infty$. The identification of redundant gauge symmetries as the ones generated by the primary[5]
65 first-class[6] constraints is based on the Dirac-Bergmann theory of constraints [34, 35], in which
66 one takes Poisson brackets of the primary first-class constraints with the fields of the theory to
67 generate gauge transformations. For details see e.g. [36–41].

68      For electromagnetism (with structure group $U(1)$) on three-dimensional space $\Sigma$, the group
69 $\mathcal{G}^I$ of boundary-preserving gauge transformations is identified as consisting of those transfor-
70 mations $g : \Sigma \to U(1)$ that become asymptotically constant [28, 30]. Furthermore, the sub-
71 group $\mathcal{G}_0^\infty$ is identified as the one generated by the Gauss law constraint, consisting of all
72 transformations $g : \Sigma \to U(1)$ that asymptotically approach the identity [28, 42]. The quo-
73 tient is then said to be isomorphic to $U(1)$ itself, i.e. the group of global (or rigid) gauge
74 symmetries [27, 28, 30, 43].

75      However, the derivations supporting these results are at the least shaky and sometimes
76 simply incorrect. The common lore is that one must impose asymptotic fall-off boundary con-
77 ditions on gauge fields, e.g.

$$A_i \to 0 + \mathcal{O}(r^{-2}), \quad i = 1, 2, 3,$$

78 to ensure finiteness of energy and/or action [44]. It is then said that the gauge group must
79 preserve these conditions [27] and must therefore be restricted to $\mathcal{G}^I$. But this argument
80 is problematic, since energy and action only depend on gauge-invariant quantities (the field
81 strength tensor). Thus there is no need to require gauge fields to become zero asymptotically:
82 we need only require that they become pure gauge, a point that was already noted by Atiyah
83 [45, Section I.4]. But any gauge transformation preserves this condition, so it would seem
84 naively that one can always allow the full gauge group $\mathcal{G}$, instead of restricting to $\mathcal{G}^I$.[7] If
85 true, this would greatly enlarge the group $\mathcal{G}_{\text{DES}}$, well beyond the group of global (rigid) gauge
86 transformations.

87      One aim of this article is to explain why we must in fact still restrict to the subgroup $\mathcal{G}^I$,
88 although this does not follow only from finiteness of energy but also from the requirement that
89 *the Lagrangian be defined on the tangent bundle to configuration space.* For finite-dimensional
90 systems this latter requirement is the foundation for proving the equivalence of the Euler-
91 Lagrange equations and stationarity of action in variational principles, see e.g. [46, Chapter
92 8] or [47, Chapter 19]. But gauge field theories are infinite-dimensional systems with infinite-
93 dimensional symmetry groups, resulting in the added difficulty that the Lagrangian is degen-
94 erate (exhibits constraints) [36, 48]. In that case, not all vectors in the tangent bundle to
95 configuration space admit solutions to the Euler-Lagrange equations of which they are the
96 initial datum [48, Section 6.4]. Still, the constraints are found in the first place through the
97 Legendre transform $L : TQ \to T^*Q$ from the tangent bundle to the cotangent bundle of the

---

[3]Hence the notation $I$, which will be used throughout to denote classes of maps that leave the asymptotic boundary conditions invariant, i.e. which are constant at infinity (except in the broken phase of the Yang-Mills-Higgs theory, where boundary-preserving transformations must actually vanish at infinity, see Section 5).

[4]Throughout this article we use the *subscript* $\infty$ to denote certain conditions at asymptotic infinity (usually the vanishing of classes of maps), which is not to be confused with the superscript denoting smoothness. Only for $\mathcal{G}_0^\infty$ have we used $\infty$ as a superscript since there we already have the subscript 0 and there is no danger of confusion.

[5]Primary constraints are constraints that are obtained without using the equations of motion.

[6]First-class constraints are constraints whose Poisson bracket with any other constraint vanishes.

[7]There is another way to formulate this critique: the very statement $A_i \to 0$ is made in a specific gauge. What we call "zero" is therefore gauge-dependent. Thus, the fact that this asymptotic boundary condition is not preserved by most gauge transformations is not surprising - it is a consequence of our working in a gauge.

configuration space $Q$. The constraint surface $\mathcal{C}$ is the image $L(TQ)$ of the tangent bundle under the Legendre transform [36, 48]. Thus, even in gauge field theories, one always starts with a Lagrangian defined on the tangent bundle to configuration space.[8]

Besides the problem of correctly identifying $\mathcal{G}^I$, there is further obscurity in the literature when $\mathcal{G}_{DES}$ is identified with the group of global (rigid) gauge symmetries. This pertains to the appropriate *rate* at which transformations $g \in \mathcal{G}^I$ must become constant asymptotically, and the rate at which elements $g \in \mathcal{G}_0^\infty$ must become the identity. It is only when these rates are exactly equal that we can conclude that the quotient of these two subgroups of $\mathcal{G}$ is isomorphic to $U(1)$ (in the Abelian case). However, in the usual approach it is not obvious that these rates are the same. To see this, note that, in 3-dimensional space, the electric field must vanish asymptotically with order at least $\mathcal{O}(r^{-3/2-\epsilon})$ to guarantee that it is square-integrable,[9] where $\epsilon > 0$ is any (small) number. As we will explain later, this same rate is needed for the gauge field itself. It is then concluded that gauge transformations $g : \Sigma \to G$ must become constant asymptotically to preserve this boundary condition. But at what rate? In the Abelian case, we would need the gauge parameter $\lambda : \Sigma \to \mathbb{R}$ to be such that its derivative $\partial_i \lambda$ falls off with order $\mathcal{O}(r^{-3/2-\epsilon})$, if it is to be boundary-preserving. But what does this imply for $\lambda$ itself? It is not obvious that we can simply conclude that $\lambda \to \text{const} + \mathcal{O}(r^{-1/2-\epsilon})$, i.e. that $\lambda$ falls off towards a constant with one power of $r$ fewer. Indeed, there are examples of functions which themselves vanish in a certain limit but whose derivative behaves very badly. Besides, as noted in [27, 28], our choice of asymptotic behavior of the fields has a large effect on what transformations $\mathcal{G}^I$ contains precisely, seemingly making the derivation of the physical gauge group quite arbitary.

Similar issues arise when considering the order of asymptotic behavior for transformations $g \in \mathcal{G}_0^\infty$. In fact, in the argument by Balachandran [42], which formed the basis for Teh's derivation [28], the requirement that $g \to 1$ asymptotically is based on the need for a certain boundary term to vanish in the calculation of a specific Poisson bracket. But this boundary term contains the electric field, and so its vanishing could also be guaranteed simply by requiring rapid enough asymptotic fall-off of the electric field, such that gauge transformations do not need to go to the identity to make sure this Poisson bracket exists. We will run into this issue again in Section 4.1 and resolve it in Section 4.3. At any rate (pun intended), it is clear that quite a lot of fine-tuning of asymptotic behavior is needed to ensure that, in the end, the quotient $\mathcal{G}_{DES} = \mathcal{G}^I / \mathcal{G}_0^\infty$ corresponds precisely to the group of global gauge transformations. This arbitrariness is highly unsatisfactory.

These ambiguities contrast sharply with other characterizations of the special status of global gauge symmetries, from which it is obvious that it is precisely the global gauge group that stands apart from other gauge transformations. We mention three such characterizations.

Firstly, in the formalization of gauge theories using fiber bundles, connections live on a principal $G$-bundle $P \to \Sigma$. Gauge transformations correspond to bundle automorphisms $P \to P$. But in the Abelian case,[10] there is clearly a special class of gauge transformations, namely the ones that are given by the global action $G : P \to P$, which forms part of the very definition of a principal bundle. A connection on $P$ is a choice of horizontal subspace at every point $p \in P$. Since the global action of $G$ on $P$ is, by definition, perfectly vertical, it is not felt by the connections.

Secondly, in the symplectic formulation of gauge theories, the global gauge group appears

---

[8]In a covariant phase space framework one may work with the first jet bundle rather than the tangent bundle. We leave a reformulation of our results into the language of jet bundles for future research.

[9]Square-integrability is required because the energy carried by the electric field is the integral of the square of its norm, and this energy is required to be finite.

[10]In the non-Abelian case the action $f : P \to P$ defined by $f(p) = ph_1$, for some $h_1 \in G$, does not necessarily define a bundle automorphism. Equivariance might fail since $f(ph_2) = ph_2 h_1$ is not necessarily the same as $f(p)h_2 = ph_1 h_2$ if $h_1, h_2$ do not commute. Yet the central elements of $G$ do define a bundle automorphism this way.

precisely as the obstruction to the possibility of a smooth symplectic reduction [2,3,8,9]. To see this, recall that any Hamiltonian group action on a symplectic manifold can be used to define a momentum map (Definition 4.1) such that, if the group acts freely[11] and properly[12] on the zero set of this momentum map, one can take a symplectic quotient [46–48]. However, since global gauge transformations can be viewed as the constant maps $g : \Sigma \to G$, they do not act freely. In the Abelian case, a connection $A$ transforms as

$$A \to A + g^{-1} dg,$$

so if $g$ is constant then $dg$ is zero, and *any* connection will be a fixed point of the global gauge group action. This prevents the possibility of a smooth symplectic reduction.[13] The symplectic quotient will instead be a stratified space [9]. In the non-Abelian case, constant gauge transformations $g$ do act by conjugation $A \mapsto g^{-1} Ag$, (non-Abelian gauge bosons are charged under the force they themselves transmit), but then the central global gauge transformations still do not act freely.

Thirdly, but relatedly, Gomes and Riello have used horizontal symplectic geometry to identify the global gauge group as carrying a different empirical status from other gauge transformations [5–9,31]. In electromagnetism this is achieved by means of the dressing

$$h[A] = \int_\Sigma \frac{dy^3}{4\pi} \frac{\partial^i A_i}{|\mathbf{x} - \mathbf{y}|},$$

which singles out the gauge-invariant component of the gauge field $A$ on 3-dimensional space $\Sigma$. This dressing corresponds to a projection onto the Coulomb gauge and is insensible precisely to the global gauge transformations, as these do not change $A$. Clearly this is related to the previous point: the common idea is that (central) global gauge transformations do not change the gauge field, whereas these do change the global phase of matter fields.[14]

Thus, we arrive at the central goal of this article: unifying the various approaches to deriving precisely the global gauge group as the one carrying DES, by carefully considering the configuration space of Yang-Mills fields and their spatial asymptotic boundary conditions. Our approach is as follows. We first construct the configuration space of gauge fields in Section 2, without working in a particular gauge. In Section 3 we then use this construction to define boundary conditions that are necessary to ensure finiteness of energy, and we examine their consequence for the structure of the configuration space of gauge fields. Subsequently, we find the redundant gauge symmetries, i.e. those generated by the Gauss law constraint, in Section 4, finally giving us the quotient of transformations with DES. Lastly, we study what happens when a Higgs field is added in Section 5, in which case we find different boundary conditions for the unbroken and broken phases.

## 2    The configuration space of gauge fields

In this Section we explain what the configuration space of Yang-Mills theories, on whose tangent bundle the Yang-Mills Lagrangian is defined, looks like. We do this without working in

---

[11]The action of a group $H$ on a set $X$ is called free if $h \cdot x = x$ for some $x \in X$ implies that $g$ is the identity.

[12]The action of a topological group $H$ (such as a Lie group) acting by homeomorphisms on a topological space $X$ (such as a manifold) is called proper if the map $H \times X \to X \times X$ is proper. A map between topological spaces is called proper if the inverse image of a compact set is itself compact.

[13]To resolve this we could consider the group $\mathcal{G}_*$ of *pointed* gauge transformations, i.e. those transformations that are the identity at some arbitary fixed point $x_0 \in \Sigma$. Then the only global transformation is the trivial one and the action of $\mathcal{G}_*$ is free, so that the symplectic reduction is a smooth space. This approach is pursued in [49]. We could also consider so-called *irreducible connections*, i.e. connections for which the holonomy group acts irreducibly. The gauge group does act freely on the space of irreducible connections [50].

[14]For this reason they are used in so-called 't Hooft beam splitter [51] constructions, see e.g. [52].

a particular gauge, which is of paramount importance for conceptual clarity. After all, if we impose boundary conditions such as $A_i \to 0$ while already working in a specific gauge, then it is not surprising that gauge transformations violate this boundary condition. However, it is then unclear whether this violation is really problematic or just an artifact of our choice to work in a gauge, and we should avoid this ambiguity.

The results of this Section are a necessary prerequisite for understanding the main point of Section 3: that the need to restrict the allowed gauge transformations to $\mathcal{G}^I$, i.e. the subgroup of transformations that leave the boundary conditions invariant, comes not directly from the boundary conditions themselves, but rather from the requirement that the domain of the Lagrangian be a tangent bundle.

Throughout this article we assume a 3+1 split of flat spacetime into $\Sigma \times \mathbb{R}$, where $\Sigma \cong \mathbb{R}^3$, and work in the temporal gauge, thus setting $A_0 = 0$. This means that we do not consider gauge transformations in the temporal component of the gauge field, but only in its spatial components. We do this because we are ultimately interested in understanding the breaking of spatial gauge transformations in the Higgs mechanism. For details on the relation between such a 3+1 split and covariant formulations of Yang-Mills theory, we refer the reader to Section 8.3 of [48] and to [53, 54].

We consider a principal $G$-bundle $P \to \Sigma$, where the structure group $G$ is some compact matrix Lie group such as $U(1)$ or $SU(N)$, with Lie algebra $\mathrm{Lie}(G) = \mathfrak{g}$. The structure group should not be confused with the gauge group $\mathcal{G} = \mathrm{Aut}(P)$ of all gauge transformations. A gauge field in Yang-Mills theory is a connection on this bundle $P$, i.e. a choice of horizontal distribution in the tangent bundle $TP$. Equivalently a gauge field can be viewed as a Lie algebra-valued 1-form on $P$, i.e. an element $A \in \Omega^1(P, \mathfrak{g})$, that is both $G$-equivariant and reproduces the Lie algebra generators of the fundamental vector fields[15] [55]. $G$-equivariance means that $r_h^* \circ A = \mathrm{Ad}_{h^{-1}} \circ A$ for all $h \in G$, where $\mathrm{Ad} : G \to \mathrm{GL}(\mathfrak{g})$ denotes the adjoint representation[16] and $r_h^* : \mathfrak{g} \to \mathfrak{g}$ the pullback of the right multiplication $r_h : G \to G$ by $h \in G$. Such a connection 1-form $A$ can be pulled down to $\Sigma$ if we have a gauge, i.e. a section $s : \Sigma \to P$, in which case it is acted upon by the gauge group $\mathcal{G}$ in the usual way:

$$ s^* \tilde{g} A = \tilde{g}^{-1} s^* A \tilde{g} + \tilde{g}^{-1} d\tilde{g}, \quad \tilde{g} \in C^\infty(\Sigma, G). $$

Here $s^* : \Omega^1(P, \mathfrak{g}) \to \Omega^1(\Sigma, \mathfrak{g})$ denotes the pullback through the gauge $s$, and we have used the isomorphism $\mathcal{G} = \mathrm{Aut}(P) \cong C^\infty(\Sigma, G)$, which sends $g \mapsto \tilde{g}$, induced by $s$.[17] Henceforth we drop the tilde on $g$, meaning that we freely switch between the gauge-invariant definition $\mathcal{G} = \mathrm{Aut}(P)$ and the gauge-dependent definition $\mathcal{G} = C^\infty(\Sigma, G)$.

If we write $\mathrm{Conn}(P)$ for the space of all connection 1-forms on $P$, then the space of "coordinates" and "velocities" of Yang-Mills theory naively consists of the tangent bundle $T\mathrm{Conn}(P)$ to $\mathrm{Conn}(P)$. However, as we will see in Section 3, asymptotic boundary conditions are required on the tangent vectors (electric fields) in this tangent bundle, thereby complicating the construction. Now, (asymptotic) boundary conditions are usually imposed on fields on space $\Sigma$ and not on the bundle $P$, so we need to bring down our fields to $\Sigma$. We could do this by working in a gauge as above, but we have just explained that it is vital to work gauge-invariantly. Fortunately, it is also possible to work gauge-invariantly on $\Sigma$, using the following.

A $k$-form $\omega \in \Omega^k(P, \mathfrak{g})$ is called *horizontal* if it vanishes whenever at least one vector it eats is vertical, i.e. if for all $p \in P$ we have $\omega_p(X_1, ..., X_k) = 0$ whenever $X_i \in V_p P = \ker(\pi_*)$ for some $1 \le i \le k$. Here $V_p P = \ker(\pi_*)$ denotes the space of vertical vectors at the point $p$, which

---

[15]That is: $A(X_\xi) = \xi$ for all $\xi \in \mathfrak{g}$, where $X_\xi$ denotes the fundamental vector field in $\mathfrak{X}(P)$ generated by $\xi$ through the right action of $G$ on $P$.

[16]Defined by $\mathrm{Ad}_h(X) = hXh^{-1}$, where $h \in G, X \in \mathfrak{g}$.

[17]The isomorphism between the two groups is as follows. If we have a $G$-valued map $g : \Sigma \to G$, then we can produce a bundle automorphism $f : P \to P$ using the section $s : \Sigma \to P$. We simply define $f(p) = p \cdot s(\pi(p))$.

should be thought of as the vectors that lie along the fibers (which are isomorphic to $G$) of $P$. Furthermore, we say a $k$-form $\omega$ is *of type Ad* if $r_h^* \circ \omega = \mathrm{Ad}_{h^{-1}} \circ \omega$ for any $h \in G$. We denote the set of horizontal $k$-forms of type Ad by $\Omega_{\mathrm{hor}}^k(P, \mathfrak{g})^{\mathrm{Ad}}$. We have the following result [55].

**Proposition 2.1** *Let $P \to \Sigma$ be a principal $G$-bundle. If $A, A' \in \Omega^1(P, \mathfrak{g})$ are two connection 1-forms then $A - A' \in \Omega_{hor}^1(P, \mathfrak{g})^{Ad}$ and for any $\omega \in \Omega_{hor}^1(P, \mathfrak{g})^{Ad}$ we have that $A + \omega$ is a connection 1-form. For the curvature we have $F(A) \in \Omega_{hor}^2(P, \mathfrak{g})^{Ad}$.*

In other words: differences of connections as well as curvatures are horizontal forms of type Ad. This is extremely useful because of the following well-known theorem [55].

**Theorem 2.2** *Let $\pi \colon P \to \Sigma$ be a principal $G$-bundle. Then $\Omega_{hor}^k(P, \mathfrak{g})^{Ad}$ and $\Omega^k(\Sigma, Ad(P))$ are canonically isomorphic as vector spaces through the pullback[18] $\pi^*$.*

Here $\mathrm{Ad}(P)$ denotes the adjoint bundle.[19] Thus, if we choose a basis connection $A_{\mathrm{ref}}$, we can view the space of connection 1-forms $\mathrm{Conn}(P)$ as the vector space $\Omega^1(\Sigma, \mathrm{Ad}(P))$. In other words: we can view differences of connections as well as curvatures as forms on $\Sigma$ instead of on $P$ *in a gauge-invariant manner*, as long as we remember that it is in reference to the basis connection $A_{\mathrm{ref}}$. For an Abelian structure group the adjoint bundle $\mathrm{Ad}(P)$ is even trivial, i.e. just $\mathrm{Ad}(P) = \Sigma \times \mathfrak{g}$, so that the space of connections becomes simply $\Omega^1(\Sigma, \mathfrak{g})$.

Now, we know what the tangent space to a vector space looks like: it is isomorphic to the original vector space, even in infinite dimensions. This allows us to obtain the tangent bundle to the space of connections. We find $T\mathrm{Conn}(P) = T\Omega^1(\Sigma, \mathrm{Ad}(P)) \cong \Omega^1(\Sigma, \mathrm{Ad}(P)) \times \Omega^1(\Sigma, \mathrm{Ad}(P))$. In electromagnetism $\mathfrak{g} = i\mathbb{R}$, so that $T\mathrm{Conn}(P)$ simply equals $\Omega^1(\Sigma) \times \Omega^1(\Sigma)$.

# 3 Asymptotic boundary conditions and the gauge group

Thus far we have not considered any boundary conditions on the connection 1-forms or the tangent vectors in, nor on the curvatures, even though this is essential for ensuring finiteness of the Lagrangian, Hamiltonian and action. The Lagrangian is an integral over $\Sigma \cong \mathbb{R}^3$, so terms that appear in the integrand must fall off asymptotically with order at least $\mathcal{O}(r^{-3-\epsilon})$ to make this integral well-defined. Let us now see what these terms are and what imposing boundary conditions implies for the group of gauge transformations $\mathcal{G}$.

## 3.1 Boundary conditions on gauge fields

Our goal is to identify a subspace $Q \subset \Omega^1(\Sigma, \mathrm{Ad}(P))$ of the space of all gauge fields which is such that the curvatures of its elements as well as the elements of its tangent bundle $TQ$ satisfy the asymptotic boundary conditions. These boundary conditions are dictated by the Lagrangian of Yang-Mills theory on $\Sigma$, which is a map $\mathcal{L} \colon TQ \to \mathbb{R}$. Elements of $TQ$ consist of pairs $(A, \alpha_A) \in Q \times T_A Q$ of gauge fields and tangent vectors. We think of $\alpha_A$ as the electric field, but it is entirely independent of $A$ as long as we do not impose the equations of motion, which is one reason why we have chosen not to use the symbol $E$ (the other reason is that we will use $E$ to denote the conjugate momentum to $A$ in Section 4). The tangent vectors $\alpha_A$ are

---

[18]Recall that for a fiber bundle $E \to N$ any map $f : M \to N$ induces a pullback bundle $f^* E \to M$. In this case the pullback (of the adjoint bundle) is the trivial vector bundle $P \times \mathfrak{g}$.

[19]The adjoint bundle is the associated real vector bundle $\mathrm{Ad}(P) = P \times_{\mathrm{Ad}} \mathfrak{g}$ constructed through the adjoint representation $\mathrm{Ad} \colon G \to \mathrm{GL}(\mathfrak{g})$. Here the product $P \times_\rho \mathfrak{g}$ signifies that we quotient $P \times \mathfrak{g}$ by the equivalence relation $(p, X) \sim (ph, \mathrm{Ad}_{h^{-1}}(X))$ for $h \in G$.

255 the "velocities" at the "coordinate" $A$. The Lagrangian of Yang-Mills theory in temporal gauge
256 is then [56]

$$\mathcal{L}(A, \alpha_A) = \frac{1}{2} \|\alpha_A\|^2 - \frac{1}{2} \|F(A)\|^2. \tag{1}$$

257 Here $F(A)$ denotes the curvature 2-form of the connection 1-form $A$, which is the magnetic
258 field, and $\|\cdot\|$ is the usual norm on forms:

$$\|\omega\|^2 = \int_\Sigma \mathrm{Tr}(\omega \wedge *\omega),$$

259 where $*$ denotes the Hodge star operator. It is important to note for our conformal analysis in
260 Section 3.2 that the hodge star operator "contains" the metric. We can derive expression (1)
261 for the Lagrangian from the usual covariant action on spacetime $M = \Sigma \times \mathbb{R}$:

$$\mathcal{S}(\tilde{A}) = -\frac{1}{2} \int_M \mathrm{Tr}\, F(\tilde{A}) \wedge *F(\tilde{A}) = -\frac{1}{2} \int_\mathbb{R} \int_\Sigma \mathrm{Tr}\, F(\tilde{A}) \wedge *F(\tilde{A}).$$

262 Here we have written $\tilde{A}$ to stress that this is a gauge field on spacetime $M$ instead of space $\Sigma$.
263 Denoting coordinates on $\Sigma$ by $x^i$ and the coordinate on $\mathbb{R}$ by $t = x^0$, it is not difficult to show
264 that the action in coordinates becomes the usual [55]

$$\mathcal{S}(\tilde{A}) = -\frac{1}{4} \int_\mathbb{R} dt \int_\Sigma dx^3\, \mathrm{Tr}(F_{\mu\nu} F^{\mu\nu}) = -\frac{1}{4} \int_\mathbb{R} dt \int_\Sigma dx^3\, \mathrm{Tr}\left(2 F_{0i} F^{0i} + F_{ij} F^{ij}\right),$$

265 where $\mu = 0, 1, 2, 3$, $i = 1, 2, 3$ and $F_{\mu\nu} = \partial_\mu A_\nu - \partial_\nu A_\mu + [A_\mu, A_\nu]$ is the antisymmetric field
266 strength tensor (which clearly satisfies $F_{00} = 0$). The term $\mathrm{Tr}(F_{0i} F^{0i})$ is (minus) the energy of
267 the electric field (the "kinetic" energy), and the term $\mathrm{Tr}(F_{ij} F^{ij})$ equals twice the energy of the
268 magnetic field (the "potential" energy).

269      If we now impose temporal gauge $A_0 = 0$ we obtain $F_{0i} = \partial_0 A_i = \dot{A}_i$. We can then rewrite
270 the action as[20]

$$\mathcal{S}(\tilde{A}) = \frac{1}{4} \int_\mathbb{R} dt \int_\Sigma dx^3\, \mathrm{Tr}\left(-2\dot{A}_i \dot{A}^i - F_{ij} F^{ij}\right) = \int_\mathbb{R} dt\, \mathcal{L}(A_i, \dot{A}_i).$$

271 But $F_{ij}$ is just the curvature of the connection $A_i$ on three-dimensional space $\Sigma$, so in coordinate-
272 free notation we find, with slight abuse of notation:

$$\mathcal{S}(\tilde{A}) = \frac{1}{2} \int_\mathbb{R} dt \int_\Sigma \mathrm{Tr}\left(\dot{A} \wedge *\dot{A} - F(A) \wedge *F(A)\right) = \frac{1}{2} \int_\mathbb{R} dt \left(\|\dot{A}\|^2 - \|F(A)\|^2\right),$$

273 where it is understood that $A \in Q \subset \Omega^1(\Sigma, \mathrm{Ad}(P))$ signifies the spatial part of $A_\mu$. Technically
274 there is no sense in which the spatial gauge field $A$ has a time-derivative $\dot{A}$. What is really meant
275 by this expression is that $\dot{A}$ should be viewed as a tangent vector at the point $A$, obtained as a
276 derivative along a curve $\mathbb{R} \to Q$. Thus we replace $\dot{A} \to \alpha_A \in T_A Q$ and we obtain the Lagrangian
277 in Eq. (1).

278      Now, we want the Lagrangian to be well-defined as an integral over $\Sigma$, and so we require
279 both $\|\alpha_A\|$ and $\|F(A)\|$ to be separately finite, anticipating also that the energy is the sum of
280 these. As these norms are just integrals over 3-dimensional space, square-integrability requires
281 that $\alpha_A$ and $F(A)$ fall-off sufficiently quickly towards spatial asymptotic infinity. Denoting by
282 $g_\Sigma$ the metric and writing

$$\omega \wedge *\omega = g_\Sigma(\omega, \omega)\, d\mathrm{Vol}_{g_\Sigma}, \quad \omega \in \Omega^k(\Sigma, \mathrm{Ad}(P)),$$

---

[20] We use $(-, +, +, +)$ signature for the metric, which explains the minus sign in $-\dot{A}_i \dot{A}^i$.

283  we need to require that, as $r \to \infty$:

285      (i) $g_\Sigma(\alpha_A, \alpha_A) \to 0 + \mathcal{O}(r^{-3-\epsilon})$;
286      (ii) $g_\Sigma(F(A), F(A)) \to 0 + \mathcal{O}(r^{-3-\epsilon})$,

288  where $\epsilon > 0$ is a small number. Since in coordinates we have $g_\Sigma(\alpha_A, \alpha_A) = g_\Sigma^{ij}(\alpha_A)_i(\alpha_A)_j$
289  and $g_\Sigma(F(A), F(A)) = g_\Sigma^{ik} g_\Sigma^{jl} F(A)_{kl} F(A)_{ij}$ (using Einstein summation convention), we find the
290  following boundary conditions in Cartesian coordinates:

292      (i) $(\alpha_A)_i \to 0 + \mathcal{O}(r^{-3/2-\epsilon})$;
293      (ii) $F(A)_{ij} \to 0 + \mathcal{O}(r^{-3/2-\epsilon})$.

295  In spherical coordinates, however, the inverse of the metric $g_\Sigma = dr^2 + r^2 d\Omega^2$ gives a fac-
296  tor $r^{-2}$ for each of the angular coordinates (and a factor $r^{-4}$ for the double angular coordinate
297  $F_{\theta\phi}$), leading to the boundary conditions:

299      (i) $(\alpha_A)_r \to 0 + \mathcal{O}(r^{-3/2-\epsilon})$,  $(\alpha_A)_\theta, (\alpha_A)_\phi \to 0 + \mathcal{O}(r^{-1/2-\epsilon})$;
300      (ii) $F(A)_{r\theta}$, $F(A)_{r\phi} \to 0 + \mathcal{O}(r^{-1/2-\epsilon})$,  $F(A)_{\theta\phi} = \mathcal{O}(r^{1/2-\epsilon})$.

302  All in all we see that the gauge field $A$ must become flat at asymptotic infinity sufficiently
303  quickly and the tangent vector "electric field" $\alpha_A$ must vanish at infinity. We note that there
304  is apparently no requirement for the gauge field itself to vanish at infinity, since it does not
305  appear in the Lagrangian directly. It only needs to become flat [45]. This raises the ques-
306  tion: do the above boundary conditions produce a proper tangent bundle $TQ$? That is: if we
307  take $Q$ to consist of those connections that become flat asymptotically at the right rate, will
308  its tangent space $T_A Q$ at a point $A \in Q$ then consist precisely of those $\alpha_A$ that approach zero
309  asymptotically at that same rate? The answer is no. To see this, we consider the space of flat
310  connections at infinity and examine its tangent space. It should consist of the zero vector only,
311  since we require $\alpha_A$ to vanish at infinity. In other words: the tangent space at infinity should
312  be 0-dimensional, which in turn implies that the space of flat connections at infinity should
313  be 0-dimensional, i.e. a discrete space. For simplicity we take it to consist of a single point,
314  i.e. some *fixed* asymptotic boundary choice of flat connection at infinity. Gauge transforma-
315  tions must leave this fixed, flat connection at infinity invariant and should therefore become
316  constant at infinity.

## 3.2  Conformal analysis

318  Let us be more precise about this. As explained in Sections 8.4 and 8.5 of [48], the con-
319  formal invariance of Yang-Mills theory allows one to make use of a conformal embedding of
320  Minkowski spacetime $(M, \eta)$ into a Lorentzian manifold $(\hat{M}, \hat{\eta})$ with compact Cauchy surfaces.
321  Such an embedding is a map $f : (M, \eta) \to (\hat{M}, \hat{\eta})$ which sends $M$ to the interior of $\hat{M}$ and which
322  is such that $f^*\hat{\eta} = K^2 \eta$ for some positive function $K$, the conformal factor. We can take $\hat{M}$ to
323  be $\mathbb{R} \times S^3$ with the metric $\hat{\eta} = -d\tau^2 + g_{S^3}$. Using standard angular coordinates $(\alpha, \beta, \gamma)$ for $S^3$
324  and spherical coordinates $(r, \theta, \phi)$ for $\mathbb{R}^3$, we have

$$g_{S^3} = d\alpha^2 + \sin^2(\alpha)\left(d\beta^2 + \sin^2(\beta)d\gamma^2\right),$$

325  and the embedding $f : \mathbb{R} \times \mathbb{R}^3 \to \mathbb{R} \times S^3$ is explicitly given by $\tau \circ f = \arctan(t+r) + \arctan(t-r)$,
326  $\alpha \circ f = \arctan(t+r) - \arctan(t-r)$, $\beta \circ f = \theta$ and $\gamma \circ f = \phi$ [48, p. 384]. This gives the

327   conformal factor

$$K^2 = \frac{4}{((t+r)^2+1)((t-r)^2+1)},$$

328   which at fixed $t$ clearly satisfies $K \to 0 + \mathcal{O}(r^{-2})$.

329       One can attach the sphere of directions at spatial infinity [2] such that $\hat{M}$ has the structure
330   of a manifold with boundary $\partial \hat{M}$ on which $K$ vanishes [48, Proposition 8.5.2]. The Cauchy
331   surface $\Sigma \cong \mathbb{R}^3$ is then mapped into the interior of a compact space $\hat{\Sigma}$ with boundary $\partial\hat{\Sigma} \cong S^2$
332   (the celestial sphere of directions at infinity), so we can view asymptotic infinity of $\Sigma$ as $S^2$.
333   For simplicity we consider $\Sigma$ at time $t = 0$, so that the conformal embedding $\Sigma \to \hat{\Sigma}$ reduces
334   to $R = 2\arctan(r), \theta = \theta, \phi = \phi$, where $0 \le R \le \pi$ denotes the radial coordinate on $\hat{\Sigma}$
335   and $\theta, \phi$ the angular coordinates on both $\Sigma$ and $\hat{\Sigma}$. It is readily verified[21] that the metric
336   $g_{\hat{\Sigma}} = dR^2 + \sin^2(R)d\Omega^2$ is pulled back to $4(1+r^2)^{-2}\left(dr^2 + r^2 d\Omega^2\right)$, i.e. the Euclidean metric
337   on $\Sigma$ with conformal factor $K = 2(1+r^2)^{-1}$. The asymptotic boundary then corresponds to
338   $R = \pi$.

339       Since gauge fields are objects that are defined without reference to the metric, they do
340   not transform with any conformal factor when considered on $\hat{\Sigma}$ [57]. Electric fields do, if
341   they are defined using the metric, transform as $E \to K^{-1}E$ (guaranteeing that there can be
342   a nonzero electric flux at infinity), though this point is a bit subtle and we will come back
343   to it in Section 4. However, component functions of forms may transform with conformal
344   factors. This is most easily seen in spherical coordinates. Indeed, let $\alpha \in \Omega^1(\Sigma, \mathrm{Ad}(P))$ and
345   write $\alpha = \alpha_r dr + \alpha_\theta d\theta + \alpha_\phi d\phi$. Suppose that $\alpha = f^*\hat{\alpha}$ for some $\hat{\alpha} \in \Omega^1(\hat{\Sigma}, \mathrm{Ad}(\hat{P}))$. Then
346   what asymptotic behavior on $\alpha$ is implied by the fact that $\hat{\alpha}$ extends smoothly to the conformal
347   boundary $\partial\hat{\Sigma}$? Writing $\hat{\alpha} = \hat{\alpha}_R dr + \hat{\alpha}_\theta d\theta + \hat{\alpha}_\phi d\phi$, it is clear that we will simply have $\hat{\alpha}_\theta = \alpha_\theta$
348   and $\hat{\alpha}_\phi = \hat{\alpha}_\phi$, since the conformal embedding $f|_\Sigma \colon \Sigma \to \hat{\Sigma}$ does not change the angular
349   coordinates. For the radial coordinate, however, we have $dR = d(2\arctan(r)) = 2dr/(1+r^2)$,
350   so that $f^*\hat{\alpha}_R = K\alpha_r$. This means that, through the conformal embedding, the radial coordinate
351   function of a 1-form is asymptotically suppressed by $r^{-2}$.

352       Since the angular coordinates are left invariant, however, the assumption that $\hat{\alpha}$ smoothly
353   extends to the boundary $\partial\hat{\Sigma}$ implies for the pulled back angular components that $\alpha_\theta, \alpha_\phi = \mathcal{O}(1)$.
354   But this is not enough for square integrability on $\Sigma$ since, as we saw before, that requires at
355   least $\alpha_\theta, \alpha_\phi = \mathcal{O}(r^{-1/2-\epsilon})$. Thus, we need to impose the additional requirement that $\hat{\alpha}|_{\partial\hat{\Sigma}} = 0$.
356   Since $\hat{\alpha}$ is smooth even on the boundary $\partial\hat{\Sigma}$, we can perform a Taylor expansion at a point on
357   the boundary in the coordinate $\rho = \pi - R \ge 0$ around $\rho = 0$. Expanding

$$\hat{\alpha}_\theta = a_0 + a_1\rho + a_2\rho^2 + ...,$$
$$\hat{\alpha}_\phi = b_0 + b_1\rho + b_2\rho^2 + ...,$$

358   we see that the requirement $\hat{\alpha}|_{\partial\hat{\Sigma}} = 0$ implies $a_0 = b_0 = 0$, so that to lowest order $\hat{\alpha}_\theta$ and $\hat{\alpha}_\phi$
359   are linear in $\rho$. But for large $r$:

$$\rho = \pi - R = \pi - 2\arctan(r) = \pi - 2\left(\frac{\pi}{2} - \frac{1}{r} + \frac{1}{3r^3} + \mathcal{O}(r^{-5})\right) = \frac{2}{r} + \mathcal{O}(r^{-3}).$$

360   Since $\hat{\alpha}_\theta, \hat{\alpha}_\phi \sim \rho$ close to the boundary $\partial\hat{\Sigma}$, this implies that $f^*\hat{\alpha}_\theta, f^*\hat{\alpha}_\phi \to 0 + \mathcal{O}(r^{-1})$. In
361   summary, the requirement that a 1-form $\alpha$ on $\Sigma$ equals the pullback of some 1-form $\hat{\alpha}$ on $\hat{\Sigma}$
362   that vanishes on the boundary, gives the conditions $\alpha_r \to 0 + \mathcal{O}(r^{-2})$ and $\alpha_\theta, \alpha_\phi \to 0 + \mathcal{O}(r^{-1})$,
363   which are strong enough to guarantee square-integrability.

---

[21]Using the relation $\sin^2(2\arctan(r)) = \left(\frac{2r}{1+r^2}\right)^2 = \frac{4r^2}{(1+r^2)^2}$.

364      In a similar fashion we can analyze what asymptotic behavior is implied for a 2-form
365  $F \in \Omega^2(\Sigma, \mathrm{Ad}(P))$ if we assume it to come from some $\hat{F} \in \Omega^2(\hat{\Sigma}, \mathrm{Ad}(\hat{P}))$ that smoothly ex-
366  tends to the boundary $\partial\hat{\Sigma}$. Since $F_{ii}$ components vanish by antisymmetricity, it suffices to
367  check $F_{r\theta}, F_{r\phi}$ and $F_{\theta\phi}$. As the radial component is the only one whose exterior derivative
368  transforms with a conformal factor (i.e. $dR = K\,dr$), we find that $f^*\hat{F}_{r\theta} = K F_{r\theta}$, and the
369  same for $F_{r\phi}$. The component $F_{\theta\phi}$ is left invariant. Thus, if a 2-form $F$ is assumed to equal a
370  pullback $f^*\hat{F}$, then we automatically find that $F_{r\theta}, F_{r\phi} \to 0 + \mathcal{O}(r^{-2})$ and $F_{\theta\phi} = \mathcal{O}(1)$. Com-
371  paring to Section 3.1 we find that this is easily enough to guarantee that $F$ is square-integrable.
372  Thus, if one assumes a 1-form $A$ on $\Sigma$ to equal the pullback of a 1-form $\hat{A}$ on $\hat{\Sigma}$, then the cur-
373  vature $F(\hat{A})$ is a well-defined 2-form on $\hat{\Sigma}$ and therefore one automatically finds that $F(A)$ is
374  square-integrable.

## 375  3.3  Boundary-preserving gauge transformations

376  Applying the conformal analysis above to a Yang-Mills field $A \in Q$ with velocity tangent vector
377  $\alpha_A \in T_A Q$ and curvature $F(A)$, we find that we need to assume that $\hat{\alpha}_{\hat{A}} \in T_{\hat{A}}\hat{Q}$ vanishes on
378  the conformal boundary $\partial\hat{\Sigma}$, whereas we do not need to assume any behavior on $F(\hat{A})$ other
379  than smoothness on $\hat{\Sigma}$. But since tangent vectors in $T_{\hat{A}}\hat{Q}$ are required to vanish on $\partial\hat{\Sigma}$, we
380  find, by the isomorphism $\hat{Q} \cong T_{\hat{A}}\hat{Q}$, that the gauge fields $\hat{A}$ must themselves vanish on $\partial\hat{\Sigma}$.
381  Another way to view this point is as follows. It can straightforwardly be seen that the space
382  of connections $\Omega^1(\partial\hat{\Sigma}, \mathrm{Ad}(\hat{P}_{\partial\hat{\Sigma}}))$ is large enough to have a tangent space which is not zero-
383  dimensional. Indeed, if we choose some connection $\hat{A}_\infty$ on $\partial\hat{\Sigma}$, then any gauge transformation
384  at infinity, i.e. an element of $\mathrm{Aut}(\hat{P}_{\partial\hat{\Sigma}})$, yields a new connection on $\partial\hat{\Sigma}$. Thus, the space of
385  connections at infinity contains at least an orbit under the action of the gauge group at infinity.
386  The tangent space to this orbit is $C^\infty(\partial\hat{\Sigma}, \mathfrak{g})$, i.e. the $\mathfrak{g}$-valued maps on $S^2$. But this tangent
387  space is clearly not 0-dimensional.

388      Thus we see that we must restrict the configuration space of Yang-Mills fields $Q$ to those
389  connections that, through the conformal embedding, equal some *fixed* choice of connection,
390  denoted $\hat{A}_\infty$, on the asymptotic boundary $\partial\hat{\Sigma}$. Then the space of connections at infinity will
391  be 0-dimensional (consisting only of this one fixed connection) and its tangent space too.
392  However, such a choice of a fixed connection at infinity obviously *breaks gauge invariance*,
393  but in a trivial sense: any gauge transformation that is not constant[22] at infinity will change
394  $\hat{A}_\infty$. In the Abelian case, the group of gauge transformations that do preserve $\hat{A}_\infty$ consists
395  precisely of all transformations that are constant at infinity. In the non-Abelian case one has
396  to take into account the fact that even constant transformations may change $\hat{A}_\infty$ by means
397  of a conjugation. The orbit of a connection under the conjugation action of the group of
398  constant gauge transformations is itself a smooth manifold (whose dimension depends on $G$),
399  with tangent vectors which are nonzero unless the boundary connection is invariant under
400  $\mathrm{Ad}(G)$. Intuitively, this corresponds to the fact that non-Abelian gauge fields carry currents
401  even in the absence of matter fields. Avoiding such currents at infinity with infinite energy
402  forces us to pick a boundary connection which is invariant under $\mathrm{Ad}(G)$, e.g. zero. Then the
403  asymptotically constant gauge group will leave this boundary choice invariant. This choice
404  of picking the zero connection at infinity so that the full asymptotically constant gauge group
405  is allowed, rather than allowing for any flat connection but only the central asymptotically
406  constant transformations, harmonizes with Doplicher-Haag-Roberts superselection theory in
407  algebraic QFT, in which the global gauge group $G$ gives rise to observable superselection sectors
408  and can in turn be reconstructed from such a superselection structure [58–63].

---

[22]One may worry that the identification of $\mathrm{Aut}(\hat{P}|_{\partial\hat{\Sigma}})$ with maps $\partial\hat{\Sigma} \to G$ fails because trivializability of $P \to \Sigma$
does not imply trivializability over the conformal boundary. However, the extension of $\hat{P} \to f(\Sigma) = \mathrm{int}(\hat{\Sigma})$ to the
boundary $\partial\hat{\Sigma}$ is a matter of choice, and we simply choose it to be trivializable.

409      In this way, we again arrive at the familiar fact that the group of boundary-preserving
410 "allowed" gauge transformations $\mathcal{G}^I$ consists of those that become constant at infinity at the
411 appropriate rate. We need not worry anymore about what this rate is precisely,[23] since it does
412 not play a role when working on the compact space $\hat{\Sigma}$, where there is only one simple condition
413 on the transformations in $\mathcal{G}^I$, namely that they are constant on $\partial\hat{\Sigma}$. It is also clear why gauge
414 fields $A$, when viewed on $\hat{\Sigma}$, automatically approach the fixed flat connection $\hat{A}_\infty \in \Omega^1(\partial\hat{\Sigma}, \mathfrak{g})$
415 at the same rate that tangent vectors $\alpha_A$ approach zero. This follows from the requirement that
416 the space of "electric fields" is precisely the tangent space to $Q$. Since $T_A Q \cong Q$, any choice of
417 asymptotic behavior for elements in $T_A Q$ automatically translates this behavior onto $Q$ itself.

# 4   Redundant gauge symmetries and constraints

419 Having reproduced the result that the subgroup of boundary-preserving gauge transformations
420 $\mathcal{G}^I$ consists of those transformations that become constant at infinity - interpreted properly as
421 the boundary of the compact space $\hat{\Sigma}$ - it is time we turn to the question of the redundancy
422 or "triviality" of these gauge transformations. That is: which elements of $\mathcal{G}^I$ are generated by
423 the Gauss law constraint, which is the primary first-class constraint of Yang-Mills theory,[24] and
424 can therefore be interpreted to be unphysical, i.e. to not have DES?

425      In constrained Hamiltonian analysis, *gauge orbits* are null directions[25] of the symplectic
426 form pulled back to the constraint surface $\mathcal{C}$ [36]. These null directions give a clear definition
427 of "gauge" in the redundant sense: they are not felt by the symplectic form, which is the central
428 object in the classical structure of the theory. It was Dirac's great insight that these gauge orbits
429 are generated by the primary first-class constraints [34]. In symplectic geometry, this idea is
430 made precise by means of the momentum map, which formalizes infinitesimally generated
431 symmetries. Indeed, in Yang-Mills theory the constraint surface equals the inverse image of
432 zero under the momentum map for the group of redundant, trivial gauge symmetries [48].
433 Thus, in order to discover precisely which transformations in $\mathcal{G}^I$ are redundant, we pursue the
434 following strategy: we calculate for which infinitesimal gauge transformations the momentum
435 map is given by the Gauss law constraint. This approach can be seen as a precise version of
436 the argument from [42] and is also pursued for finite boundaries in [9], and we will follow
437 it now to highlight how local and global gauge symmetries obtain a different physical status:
438 only the former have the Gauss law constraint as their momentum map and should therefore
439 be viewed as redundant.

440      However, we will then explain the weakness of such an approach in the setting of asymp-
441 totic boundaries: we run into the same issues about the appropriate rates of asymptotic be-
442 havior as those highlighted in the introduction. Thus we will be forced to revert back to
443 the compact space $\hat{\Sigma}$ related to $\Sigma$ through a conformal embedding. We will then see that
444 the redundant gauge transformations are the ones that equal the identity on the conformal
445 boundary $\partial\hat{\Sigma}$, though understanding the behavior of the electric field on $\partial\hat{\Sigma}$ is a bit tricky.
446 Subsequently we generalize this result by explaining the notion of an *infinitesimal localizable*
447 *symmetry*, which in the mathematical literature are the symmetries that yield Noether's second
448 theorem and the resulting constraints, and are therefore redundant. Only global gauge sym-
449 metries are not localizable, so these should be viewed as carrying a different empirical status
450 than local gauge symmetries. They are symmetries that do not lead to constraints, similar to

---

[23]But note that, by choosing to formalize asymptotic infinite in the way done here, we picked $\mathcal{O}(r^{-2})$ as the preferred fall-off rate.

[24]Besides the $\Pi^0 = 0$ constraint that tells us that the time-component $A_0$ of the gauge field is a Lagrange multiplier, but which is excluded in our analysis because we are working in temporal gauge $A_0 = 0$ from the beginning.

[25]A symplectic form is required to be non-degenerate only on the full phase space and not on the constraint surface.

451 e.g. rotational symmetry for a point particle moving in Euclidean space.[26]

## 4.1 The momentum map for the gauge group

453 Let $Q_{A_\infty} \subset \Omega^1(\Sigma, \text{Ad}(P))$ denote the space of connections on $P \to \Sigma$ satisfying the boundary
454 conditions we arrived at in the previous Section, i.e. approaching a fixed flat[27] connection $A_\infty$
455 invariant under $\text{Ad}(G)$ at infinity at the right rate. Let us, for simplicity, assume $P$ has now
456 been trivialized, i.e. that we work in a specific gauge. Then $\text{Ad}(P) = P \times_{\text{Ad}} \mathfrak{g} \cong \Sigma \times \mathfrak{g}$, so that
457 $Q_{A_\infty} \subset \Omega^1(\Sigma, \mathfrak{g})$. To study the momentum map for Yang-Mills theory, we need to know what the
458 phase space looks like. In Section 2 we already found the domain of the Lagrangian, namely
459 the tangent bundle to configuration space $TQ \cong Q_{A_\infty} \times Q_\infty$ (the subscript for $Q_\infty$ serves to
460 remind us that these 1-forms vanish asymptotically). The phase space is a dense subspace[28]
461 $\mathcal{P} := Q_{A_\infty} \times \Omega^2_\infty(\Sigma, \mathfrak{g}) \subset T^*Q_{A_\infty}$ of the cotangent bundle [46]. It consists of pairs $(A, E)$ with
462 $A \in Q_{A_\infty}$ and $E \in \Omega^2_\infty(\Sigma, \mathfrak{g}) \subset T^*_A Q_{A_\infty}$. Here $\Omega^2_\infty(\Sigma, \mathfrak{g})$ denotes the space of 2-forms that
463 approach zero asymptotically at the appropriate rate to be square-integrable. These 2-forms
464 can indeed be viewed as elements of the cotangent space $T^*_A Q_{A_\infty}$, which consists of covectors
465 $T_A Q_{A_\infty} \to \mathbb{R}$, through their action on an element $\alpha_A \in T_A Q_{A_\infty} \cong Q_\infty \subset \Omega^1(\Sigma, \mathfrak{g})$ by means of
466 the conjugate pairing [56]

$$E(\alpha_A) = \langle \alpha_A, E \rangle = \int_\Sigma \text{Tr}\ (\alpha_A \wedge E). \tag{2}$$

467 The constraint for Yang-Mills theory is the Gauss law [64]

$$D_A E := dE + [A \wedge E] = 0.$$

468 The action of the boundary-preserving gauge group $\mathcal{G}^I$ lifts to phase space in the obvious way:

$$\forall g \in \mathcal{G}^I: \qquad g \cdot (A, E) = (g^{-1}Ag + g^{-1}dg, g^{-1}Eg).$$

469 The Lie algebra $\text{Lie}(\mathcal{G}^I)$ is isomorphic to $C^\infty_I(\Sigma, \mathfrak{g})$, i.e. the space of smooth gauge transfor-
470 mation parameters that leave the boundary conditions invariant by becoming constant to-
471 wards infinity at the right rate (we recall that this rate is determined by the conformal fac-
472 tor from the previous Section). We equip $\mathcal{P} \subset T^*Q_{A_\infty}$ with the canonical symplectic form
473 $\omega = \int_\Sigma \text{Tr}\ \mathbb{d}A \wedge \mathbb{d}E$, where the $\mathbb{d}$ symbol is used to stress that this is the derivative opera-
474 tor on the infinite-dimensional phase space of fields and not the $d$ on 3-space $\Sigma$. Henceforth
475 we will occasionally use double-slashed symbols to stress that these objects are defined on
476 infinite-dimensional phase space $\mathcal{P}$.

477 We should like to check that, with this symplectic form, the Gauss law constraint generates
478 gauge transformations, i.e. check for which gauge parameters $\xi \in \text{Lie}(\mathcal{G}^I)$ the momentum
479 map equals the Gauss law. Let us recall the definition of the momentum map [47].

---

[26]The Lagrangian for such a particle is $\mathcal{L}(q, v) = \frac{1}{2}g_q(v, v) - V(q)$, where $g_q: T_q\mathbb{R}^3 \times T_q\mathbb{R}^3 \to \mathbb{R}$ is a metric. The symmetries of the system are the isometries that leave the potential $V$ invariant. If the potential is rotationally symmetric, then rotations are symmetries. But the Legendre transform $L: T\mathbb{R}^3 \to T^*\mathbb{R}^3$ is given by $v_q \to g_q(v, \cdot)$, which is clearly a diffeomorphism. This means that there are no constraints.

[27]Though, as we have seen, on $\hat{\Sigma}$ this flatness need not be explicitly demanded!

[28]The full cotangent bundle would include distribution-like functionals that are not smooth and which we want to exclude. One could of course also consider restricting $Q_{A_\infty}$ further and allow for the full cotangent bundle $T^*Q$. For instance, one could considering taking $Q$ to consist of only Schwarz functions, so that $T^*Q$ consists of tempered distributions. The power of our argument in this article is that such alterations would not change the main result that the asymptotic symmetry group is the global gauge group.

**Definition 4.1** *Let $(\mathcal{P}, \omega)$ be a symplectic manifold and $H$ a Lie group that acts on $\mathcal{P}$ by symplectomorphisms.*[29] *Let $\mathfrak{h}$ denote the Lie algebra of $H$ with dual $\mathfrak{h}^*$, and write $\langle \cdot, \cdot \rangle \colon \mathfrak{h}^* \times \mathfrak{h} \to \mathbb{R}$ for the pairing of the algebra and its dual. Then a momentum map for the $H$-action on $\mathcal{P}$ is an equivariant*[30] *map $\mu \colon \mathcal{P} \to \mathfrak{h}^*$ such that, for all $\xi \in \mathfrak{h}$, we have:*

$$\mathrm{d}\langle \mu, \xi \rangle = \iota_{\mathbb{X}_\xi} \omega = \omega(\mathbb{X}_\xi, \cdot).$$

*Here, $\mathbb{X}_\xi$ denotes the fundamental vector field*[31] *generated by $\xi$, and $\langle \mu, \xi \rangle$ is understood as a function $\langle \mu, \xi \rangle \colon \mathcal{P} \to \mathbb{R}$, defined as follows: $\langle \mu, \xi \rangle(x) = \langle \mu(x), \xi \rangle$.*

The idea behind this definition is that the fundamental vector field $\mathbb{X}_\xi$ infinitesimally generates the $H$-action with parameter $\xi$, while the values $\langle \mu, \xi \rangle$ of the momentum map for specific $\xi$ provide constants of motion. The required relation $\mathrm{d}\langle \mu, \xi \rangle = \omega(\mathbb{X}_\xi, \cdot)$ can then be viewed in the light of Noether's theorem: it relates the conservation of the constants of motion to the symmetry of the theory.[32] For gauge symmetries it is a version of the generation of gauge symmetries by taking Poisson brackets of fields with the smeared Gauss law.

We will now check that the Gauss law constraint is indeed the momentum map for the gauge group. We find that, by partial integration, the smeared Gauss constraint splits into a "bulk" term corresponding to the infinitesimally generated gauge symmetries and a boundary term (the electric flux). The boundary term must vanish, leading to a condition on the gauge transformation parameters. We will present our derivation on the Cauchy surface $\Sigma \cong \mathbb{R}^3$, interpreting $\partial \Sigma$ as an asymptotic boundary, but the exact same derivation would work on the compact space $\hat{\Sigma}$ with actual boundary $\partial \hat{\Sigma}$, as long as one takes care of the appropriate conformal factors that appear in the integrals. We will comment more on the conformal behavior of the electric field and the electric flux in Section 4.2.

The momentum map $\mu \colon \mathcal{P} \to \Omega^3_I(\Sigma, \mathfrak{g}) \subset \mathrm{Lie}(\mathcal{G}^I)^*$, where the $I$ denotes appropriate asymptotic fall-off behavior, for the action of the gauge group $\mathcal{G}^I$ on $\mathcal{P} \subset T^* Q_{A_\infty}$ is supposed to be the Gauss law[33] constraint $\mu(A, E) = D_A E$ [48, 64]. Here we identify $\eta \in \Omega^3_I(\Sigma, \mathfrak{g})$ as an element in the dual $C^\infty_I(\Sigma, \mathfrak{g})^*$, through the pairing $\langle \eta, \xi \rangle = \int_\Sigma \mathrm{Tr}(\xi \wedge \eta)$, similar to the pairing defined earlier. Thus, for any $\xi \in C^\infty_I(\Sigma, \mathfrak{g})$ (and using Stokes' theorem/partial integration), we have:

$$
\begin{aligned}
\langle \mu, \xi \rangle(A, E) &= \int_\Sigma \mathrm{Tr}\, D_A E \wedge \xi = \int_\Sigma \mathrm{Tr}\,(dE + [A, E]) \wedge \xi = \int_\Sigma \mathrm{Tr}\,(dE \wedge \xi - [E, A] \wedge \xi) \\
&= -\int_\Sigma \mathrm{Tr}\, E \wedge d\xi + \int_{\partial \Sigma} \mathrm{Tr}\, E \wedge \xi - \int_\Sigma \mathrm{Tr}\, E \wedge [A, \xi] = -\int_\Sigma \mathrm{Tr}\, E \wedge D_A \xi + \int_{\partial \Sigma} \mathrm{Tr}\, E \wedge \xi,
\end{aligned}
\tag{3}
$$

where we have used the ad-invariance of the trace, i.e. $\mathrm{Tr}([E, A] \wedge \xi) = \mathrm{Tr}(E \wedge [A, \xi])$. Note that for consistency we have used the $\wedge$ symbol even on $\xi \in C^\infty_I(\Sigma, \mathfrak{g})$, even though it is a 0-form.

But, if $\mu(A, E) = D_A E$ really is to define the momentum map for the action of $\mathcal{G}^I$, then by definition it must satisfy the property

$$\mathrm{d}\langle \mu, \xi \rangle = \iota_{\mathbb{X}_\xi} \omega := \omega(\mathbb{X}_\xi, \cdot), \qquad \xi \in C^\infty_I(\Sigma, \mathfrak{g}), \tag{4}$$

---

[29]I.e. the action of $H$ preserves the symplectic form $\omega$.

[30]With respect to the $H$-action on $\mathcal{P}$ and the coadjoint action on $\mathfrak{h}^*$.

[31]In this definition we use the double-slashed notation because this agrees with our subsequent calculations, but of course this definition of the momentum map is also valid for finite-dimensional symplectic manifolds.

[32]For technical details see [46–48], for a conceptual exposition see [65].

[33]If we consider Maxwell theory, then the momentum map $\mu$ applied to an element $\xi \in C^\infty_I(\Sigma, \mathfrak{g})$ is just the familiar Gauss law $\nabla \cdot \mathbf{E}$ smeared with $\xi$. This can be seen by switching to the physicists' convention $\xi = i\lambda$ and writing $D_A E = \nabla \cdot \mathbf{E}$, yielding $i \int_\Sigma d^3 x\, \lambda(\mathbf{x}) \nabla \cdot \mathbf{E}(\mathbf{x})$.

511   where $\mathbb{X}_\xi \in \mathfrak{X}(\mathcal{P})$ denotes the fundamental vector field on $\mathcal{P}$ generated by the Lie algebra
512   element $\xi$. We will now check what assumption on the asymptotic behavior of the gauge
513   transformation parameter is required for the above condition to hold.

514        To this end we first calculate the right- and left-hand sides of Eq. (4) separately and then
515   compare them. We begin with the right-hand side, i.e. $\omega(\mathbb{X}_\xi, \cdot)$. By definition, for any function
516   $\mathbb{F} \in C^\infty(\mathcal{P})$, we have:

$$\mathbb{X}_\xi(\mathbb{F})(A, E) = \frac{d}{dt}\bigg|_{t=0} \mathbb{F}\left(e^{t\xi} \cdot (A, E)\right) = \frac{d}{dt}\bigg|_{t=0} \mathbb{F}\left(e^{-t\xi} A e^{t\xi} + e^{-t\xi} d(e^{t\xi}), e^{-t\xi} E e^{t\xi}\right).$$

517   For the functions $\mathbb{F} = A$ and $E$, this simply gives:

$$\mathbb{X}_\xi(A) = \frac{d}{dt}\bigg|_{t=0} \left(e^{-t\xi} A e^{t\xi} + e^{-t\xi} d(e^{t\xi})\right) = -\xi A + A\xi + d\xi = [A, \xi] + d\xi = D_A \xi,$$

$$\mathbb{X}_\xi(E) = \frac{d}{dt}\bigg|_{t=0} \left(e^{-t\xi} E e^{t\xi}\right) = -\xi E + E\xi = [E, \xi].$$

518   Thus if we put $\mathbb{X}_\xi$ in the first slot of the symplectic form $\omega = \int_\Sigma \mathrm{Tr}\, \mathrm{d}A \wedge \mathrm{d}E$, i.e. the right-hand
519   side of Eq. (4), we get:

$$\omega_{(A,E)}(\mathbb{X}_\xi, \cdot) = \int_\Sigma \mathrm{Tr}\, \left(\mathrm{d}A(\mathbb{X}_\xi) \wedge \mathrm{d}E - \mathrm{d}E(\mathbb{X}_\xi) \wedge \mathrm{d}A\right) = \int_\Sigma \mathrm{Tr}\, \left(\mathbb{X}_\xi(A) \wedge \mathrm{d}E - \mathbb{X}_\xi(E) \wedge \mathrm{d}A\right)$$

$$= \int_\Sigma \mathrm{Tr}\, \left(([A, \xi] + d\xi) \wedge \mathrm{d}E - [E, \xi] \wedge \mathrm{d}A\right) = \int_\Sigma \mathrm{Tr}\, \left(D_A\xi \wedge \mathrm{d}E - [E, \xi] \wedge \mathrm{d}A\right).$$

$$(5)$$

520   The left hand-side of Eq. (4) gives:

$$\mathrm{d}\langle \mu, \xi \rangle = \mathrm{d} \int_\Sigma \mathrm{Tr}\, D_A E \wedge \xi = \int_\Sigma \mathrm{Tr}\, \left(\mathrm{d}(D_A E) \wedge \xi - D_A E \wedge \mathrm{d}\xi\right).$$

521   However, we cannot immediately see how this agrees with the expression in Eq. (5), because
522   Eq. (5) contains a term linear in $D_A\xi$, while the above result has a term that is linear in $\xi$.
523   Thus we need to do the partial integration in Eq. (3), which gives:

$$\mathrm{d}\langle \mu, \xi \rangle = - \int_\Sigma \mathrm{Tr}\, \left(\mathrm{d}E \wedge D_A\xi - E \wedge \mathrm{d}(D_A\xi)\right) + \mathrm{d} \int_{\partial\Sigma} \mathrm{Tr}\, E \wedge \xi. \qquad (6)$$

524   The second term in the first integral can be rewritten as:

$$E \wedge \mathrm{d}(D_A\xi) = E \wedge \mathrm{d}(d\xi + [A, \xi]) = E \wedge \mathrm{d}[A, \xi] = E \wedge [\mathrm{d}A, \xi] = E \wedge \mathrm{d}A\xi - E \wedge \xi \mathrm{d}A$$

$$= -E\xi \wedge \mathrm{d}A + \xi E \wedge \mathrm{d}A - \xi E \wedge \mathrm{d}A + E \wedge \mathrm{d}A\xi = -[E, \xi] \wedge \mathrm{d}A + [E \wedge \mathrm{d}A, \xi].$$

525   Thus, the first integral in Eq. (6) equals:

$$\int_\Sigma \mathrm{Tr}\, \left(D_A\xi \wedge \mathrm{d}E + E \wedge \mathrm{d}(D_A\xi)\right) = \int_\Sigma \mathrm{Tr}\, \left(D_A\xi \wedge \mathrm{d}E - [E, \xi] \wedge \mathrm{d}A + [E \wedge \mathrm{d}A, \xi]\right).$$

526   But the trace of the full commutator term gives zero,[34] so we obtain precisely the final ex-
527   pression in Eq. (5)! This implies that from requiring that the Gauss constraint $\mu(A, E) = D_A E$
528   is the momentum map for the action of the gauge group, it follows that the boundary term

---

[34]Or, since the trace is ad-invariant, we could also immediately have rewritten $\mathrm{Tr}\, E \wedge [\mathrm{d}A, \xi] = -\mathrm{Tr}\, [E, \xi] \wedge \mathrm{d}A$.

529 in Eq. (6) must be zero. Since that boundary term is an integral of $E \wedge \xi$, the condition that
530 needs to be imposed on $\xi$ to ensure that this term vanishes depends on the asymptotic fall-off
531 behavior of $E$. Heuristically speaking, if we take $E \to 0 + \mathcal{O}(r^{-2})$, then $\xi$ just needs to go to
532 zero without any particular fall-off rate (since the integral is over a 2-sphere with radius sent
533 to infinity, which grows as $r^2$). If instead we demand $E \to 0 + \mathcal{O}(r^{-3/2-\epsilon})$, then we must have
534 $\xi \to 0 + \mathcal{O}(r^{-1/2})$ to guarantee that there is no boundary term. However, if we require only
535 slightly stronger asymptotic fall-off conditions on the electric field, e.g. $E \to 0 + \mathcal{O}(r^{-2-\epsilon})$,
536 then apparently there no longer is any need for asymptotic requirements on $\xi$. Thus this ap-
537 proach does not provide a completely unambiguous and satisfactory answer to the question
538 of precisely what asymptotic behavior of gauge transformations is required to be able to call
539 them redundant. Moreover, even if we do conclude that we must have e.g. $\xi \to 0 + \mathcal{O}(r^{-1/2})$,
540 it is not yet clear that the quotient $\mathcal{G}^I / \mathcal{G}_0^\infty$ will be precisely the group of global gauge trans-
541 formations, even though this global group can be pristinely deduced from other approaches,
542 as was explained in Section 1. We will definitively resolve this issue in Section 4.3.

## 4.2 Electric flux through infinity

544 Before we do this, however, let us reconsider the case in which we require asymptotic fall-off
545 behavior of order $\mathcal{O}(r^{-2})$ on both the gauge and electric fields. As we saw in Section 3.2, this
546 specific behavior can be formalized by means of a conformal compactification of Minkowski
547 spacetime with conformal factor $K \sim r^{-2}$. But how does the electric field $E$ transform under
548 this conformal compactification? At the beginning of this Section, we defined electric fields
549 in the Hamiltonian picture as 2-forms in $\Omega^2_\infty(\Sigma, \mathfrak{g}) \subset T^* Q_{A_\infty}$ which act on tangent vectors
550 $\alpha_A \in T_A Q_{A_\infty}$ through the canonical pairing in Eq. (2). Since 2-forms are not defined with
551 reference to the metric, electric fields $E$ do not transform with any conformal factor under the
552 conformal embedding $\Sigma \to \hat{\Sigma}$ [2]. Thus it seems that, similar to the "electric fields" $\alpha_A$, the
553 asymptotic fall-off requirement $E \to 0 + \mathcal{O}(r^{-2})$ simply translates to the requirement that $\hat{E}$
554 vanishes on $\partial \hat{\Sigma}$. But this raises a paradox: the integral $\int_\Sigma \mathrm{Tr}(D_A E)$ equals the electric flux,
555 which can be nonzero since by Gauss's theorem this integral is just a limit of an integral over
556 a sphere of radius $r$ with $r \to \infty$. This integral grows with $r^2$, canceling the fall-off behavior
557 $\mathcal{O}(r^{-2})$. On the other hand, if $\hat{E}$ is zero on $\partial \hat{\Sigma}$, then the flux

$$\int_{\hat{\Sigma}} \mathrm{Tr}\left(D_{\hat{A}} \hat{E}\right) = \int_{\partial \hat{\Sigma}} \mathrm{Tr}\left(\hat{E}\right)$$

558 vanishes. But the two calculations of the electric flux are supposed to agree.

559 The problem with this line of thinking lies in the precise definition of fall-off behavior for
560 forms. The notation $\mathcal{O}(r^{-2})$ only makes sense for functions, which can be obtained from forms
561 by feeding them vector fields. But if $E$ is a 2-form, then it needs to be fed two vector fields
562 to produce a function, so in coordinates it has two indices. This does not coincide with our
563 intuition of the electric field as a vector field $\mathbf{E}$ with three components $E^i$, each of which must
564 fall off asymptotically with order at least $r^{-3/2-\epsilon}$ to ensure that $\|\mathbf{E}\|^2$ is integrable. Instead,
565 it is more sensible to use the hodge star involution to write $E = *\mathcal{E}$, where $\mathcal{E} \in \Omega^1(\Sigma, \mathfrak{g})$ is a
566 1-form, and to require $\mathcal{E}_i \to 0 + \mathcal{O}(r^{-2})$. The Gauss law then becomes $D_A * \mathcal{E} = 0$ [2]. The
567 1-form $\mathcal{E}$ exhibits the exact same conformal behavior as $\alpha_A$, meaning that $\hat{\mathcal{E}}$ must vanish on $\partial \hat{\Sigma}$
568 to ensure the square-integrability of its pullback to $\Sigma$. The 2-form $E = *\mathcal{E}$, however, is defined
569 through the hodge star in terms of the metric, and therefore it does pick up[35] a conformal
570 factor: $\hat{E} = K^{-1} E$ [2]. This ensures that $\hat{E}$ can be any smooth function on $\hat{\Sigma}$ (not necessarily
571 vanishing on the boundary), so that there can indeed be a nonzero electric flux through infinity.

---

[35]To see this, consider the simpler example of the function 1 which is identically 1 on $\Sigma$. This function does not
pick up any conformal factor. But $d\mathrm{Vol} = *1$ does, since in coordinates it is the square root of the determinant of
the metric.

572    With this complication clarified, there is a useful conclusion to be drawn from the deriva-
573 tion in Section 4.1. By partial integration the momentum map naturally falls into two parts,
574 i.e. two integrals, viz. $\int_\Sigma \mathrm{Tr}\, E \wedge D_A \xi$ and the boundary term $\int_{\partial\Sigma} \mathrm{Tr}\, E \wedge \xi$. The first corre-
575 sponds precisely to the symmetries that are infinitesimally generated by the fundamental vec-
576 tor fields $\mathbb{X}_\xi$, whereas the second does not [66] - it equals the smeared electric flux through
577 infinity [9]. Symmetries generated by the fundamental vector fields $\mathbb{X}_\xi$ therefore satisfy the
578 requirement that $\xi$ vanishes asymptotically, i.e. that $\hat{\xi}|_{\partial\hat{\Sigma}} = 0$. This means that only gauge
579 transformations vanishing at infinity, i.e. local transformations, are associated to the Gauss
580 law constraint through Noether's second theorem [48, Proposition 7.2.6]. Global gauge trans-
581 formations, which do act non-trivially at infinity, are not included and only appear in Noether's
582 first theorem [67].

### 583 4.3 Infinitesimal localizable symmetries

584 In the conformal picture, the boundary-preserving gauge transformations are constant on $\partial\hat{\Sigma}$
585 and the unphysical transformations vanish on $\partial\hat{\Sigma}$. The physical gauge group therefore equals a
586 copy of $G$ on the boundary, viewed as the global gauge group of constant transformations. But
587 a weakness of this conclusion is that it seems to depend on the choice of asymptotic boundary
588 conditions for the fields. If we really want the weakest possible conditions that still ensure
589 a finite Lagrangian, i.e. fall-off with order $\mathcal{O}(r^{-3/2-\epsilon})$ on $\alpha_A$ (and therefore $A$ itself) and on
590 $\mathcal{E} = *E$, then the conformal analysis is inapplicable because the conformal factor $K \sim r^{-2}$
591 suppresses the radial components of the 1-forms too strongly. This issue can be remedied,
592 however, by noting that gauge parameters $\xi$ which preserve $\mathcal{O}(r^{-3/2-\epsilon})$ fall-off, i.e. which are
593 such that $\partial_i \xi \to 0 + \mathcal{O}(r^{-3/2-\epsilon})$, must satisfy $\xi \to \mathrm{const} + \mathcal{O}(r^{-1/2-\epsilon})$ (although the converse
594 does not hold, think e.g. of $\sin(r)/r$). Indeed, taking $\xi$ to be real-valued for simplicity (but
595 with obvious generalization to cases other than $G = U(1)$), by the fundamental theorem of
596 calculus:

$$\left| \lim_{s\to\infty} \xi(s,\theta,\phi) - \xi(r,\theta,\phi) \right| = \left| \int_r^\infty \partial_s \xi(s,\theta,\phi) ds \right| \leq \int_r^\infty C s^{-3/2-\epsilon} ds \leq 2C r^{-1/2-\epsilon}.$$

597 From this it follows that $\lim_{s\to\infty} \xi(s,\theta,\phi) =: c_{\theta,\phi}$ exists and that $|\xi(r,\theta,\phi) - c_{\theta,\phi}| = \mathcal{O}(r^{-1/2-\epsilon})$.
598 A priori it could be the case that the constant $c_{\theta,\phi}$ is different in each angular direction, but it is
599 straightforward to show that this cannot be so. At fixed $r$ we can connect any two points $x, y$
600 by a curve $\gamma$ of length at most $\pi r$. This factor of $r$ is cancelled by the $1/r$ that appears in the
601 angular components of the gradient $\nabla \xi$ expressed in spherical coordinates. Integrating over
602 $\gamma$ we find that $|\xi(x) - \xi(y)| \leq \pi \tilde{C} r^{-1/2-\epsilon}$. This shows that $c_{\theta,\phi} = c_{\theta',\phi'}$. Thus we find that any
603 gauge parameter $\xi$ whose derivative $d\xi$ is square-integrable satisfies $\xi \to \mathrm{const} + \mathcal{O}(r^{-1/2-\epsilon})$.
604 But the unphysical gauge transformations generated by the Gauss constraint are precisely the
605 subset of these gauge parameters for which the constant at infinity equals zero. Thus the
606 quotient of these two algebras of gauge transformations is just $\mathfrak{g} = \mathrm{Lie}(G)$, which when expo-
607 nentiated gives the global gauge group.

608    If, on the other hand, we choose stronger conditions, e.g. $E \to 0 + \mathcal{O}(r^{-3})$ or that $E$ is
609 a Schwarz function, then the boundary term in Eq. (3) automatically vanishes, regardless
610 of the asymptotic behavior of the gauge parameter $\xi$, obviating the need for the condition
611 $\xi|_{\partial\hat{\Sigma}} = 0$. The goal of the remainder of this Section is to explain why global gauge symmetries
612 nonetheless *never* play a role in Noether's second theorem, i.e. why they do not give rise to
613 constraints, and should therefore not be considered to be redundant even if the boundary term
614 in Eq. (3) vanishes due to stricter boundary conditions.

615    In the mathematical physics literature the symmetries that give rise to constraints through
616 Noether's second theorem are the so-called *infinitesimal localizable symmetries*. These form an

617   ideal (under the Lie bracket) $\mathfrak{G} \subset \text{Lie}(\mathcal{G}^I)$ of the Lie algebra of the full boundary-preserving
618   symmetry group, and the constraint surface is the zero locus of the momentum map for the
619   infinitesimal localizable symmetries (see Section 7.5 of [48] or [3, 9]), i.e.

$$\mathcal{C} = \mu_{\mathfrak{G}}^{-1}(0).$$

620   For this reason the infinitesimal localizable symmetries should be identified as the redundant,
621   "trivial" ones. When exponentiated, they generate the minimal symmetry group that must
622   be called *gauge* in the sense of "unphysical" in order to guarantee an appropriate form of
623   determinism. These infinitesimal localizable symmetries are introduced in Definition 7.2.5
624   of [48], but we will not reproduce that definition here, since it is based on the formalism of
625   jet bundles. However, when adapted to our case at hand, it reads as follows [3]:

626   **Definition 4.2** *An infinitesimal symmetry $\xi \in \text{Lie}(\mathcal{G}^I) = C_I^\infty(\Sigma, \mathfrak{g})$ is called localizable if it*
627   *vanishes on the (asymptotic) boundary of $\Sigma$ and if for any pair of open sets $U, V \subset \Sigma$ with*
628   *disjoint closures, there exists a $\xi' \in \text{Lie}(\mathcal{G}^I)$ such that*

$$\begin{aligned}
\xi(x) &= \xi'(x), \quad x \in U; \\
\xi'(x) &= 0, \qquad x \in V.
\end{aligned}$$

629   In other words: an infinitesimal symmetry is localizable if it is zero at asymptotic infinity and
630   for any two disjoint open regions we can always find another infinitesimal symmetry that is
631   equal to the original one on the one region, but zero on the other. That is: we can always
632   localize the infinitesimal symmetry to some open region of space.

633       Clearly, global gauge transformations are not localizable since they do not vanish at asymp-
634   totic infinity, or more precisely, at the boundary $\partial\hat{\Sigma}$ of the compactified space $\hat{\Sigma}$ from Section
635   3. The question, then, is whether *all other* infinitesimal symmetries in $\text{Lie}(\mathcal{G}^I)$ are localizable.
636   If this is so, then the quotient $\mathcal{G}_0^I/\mathcal{G}_0^\infty$ equals precisely the global gauge group $G$, where $\mathcal{G}_0^I$ is
637   the identity component of $\mathcal{G}^I$ and $\mathcal{G}_0^\infty$ denotes the group generated by all $e^\xi$ with $\xi \in \mathfrak{G}$.

638       Let us therefore check that all gauge symmetries except the global ones are localizable.
639   This is done most easily by working on $\hat{\Sigma}$. There $\text{Lie}(\mathcal{G}^I)$ consists of all maps $\hat{\xi}: \hat{\Sigma} \to \mathfrak{g}$ that
640   are constant on $\partial\hat{\Sigma}$. We note that $\text{Lie}(\mathcal{G}^I)/\mathfrak{g}$, with $\mathfrak{g}$ viewed as the constant maps in $C_I^\infty(\Sigma, \mathfrak{g})$,
641   is isomorphic to the algebra $\mathfrak{G}_\infty$ of all maps $\hat{\xi}: \hat{\Sigma} \to \mathfrak{g}$ that vanish on $\partial\hat{\Sigma}$. We need to show
642   that $\mathfrak{G}_\infty = \mathfrak{G}$, i.e. that the algebra of infinitesimal localizable symmetry consists precisely of
643   the gauge parameters that vanish on $\partial\hat{\Sigma}$.

644       There are two situations to consider: if $U, V$ are the open subsets from the above definition,
645   such that $\hat{\xi} \in \text{Lie}(\mathcal{G}^I)$ must be localized on $U$ relative to $V$, then either $U$ could lie in the interior
646   of $\hat{\Sigma}$ or contain (part of) the boundary $\partial\hat{\Sigma}$. In the first case it is obvious that $\hat{\xi}$ can be localized:
647   we just use a $\mathfrak{g}$-valued bump function $\hat{f}$ that is the identity on $U$ and becomes zero very quickly
648   outside of $U$, in particular on $V$. It is then clear that $\hat{f} \cdot \hat{\xi}$ will be the required element of $\text{Lie}(\mathcal{G}^I)$
649   that agrees with $\hat{\xi}$ on $U$ and is zero on $V$. In the case in which $U$ contains part of the boundary
650   it is not immediately clear whether $\hat{f} \cdot \hat{\xi} \in \text{Lie}(\mathcal{G}^I)$. But since $\hat{\xi}$ is zero on $\partial\hat{\Sigma}$, so is the
651   product $\hat{f} \cdot \hat{\xi}$. This means $\hat{f} \cdot \hat{\xi} \in \mathfrak{G}_\infty \subset \text{Lie}(\mathcal{G}^I)$. We conclude that the algebra of infinitesimal
652   localizable symmetries $\mathfrak{G}$ is indeed $\mathfrak{G}_\infty$. Thus we find that, in Yang-Mills theory, localizability
653   effectively reduces to just the condition of vanishing at infinity.[36] Of course, this is not a
654   surprising result, since gauge symmetries are meant to be localizable. But we clearly see that
655   if a field theory contains only global (rigid) symmetries, then no infinitesimal transformation
656   is ever localizable, in which case $\mathfrak{G}$ would be zero and there would be no constraints.

---

[36]Note that this result is quite independent of the precise form of $Q$ and $\mathcal{G}^I$. No matter what asymptotic conditions on the fields are required, we always find that the redundant gauge transformations are all elements of $\mathcal{G}^I$ which vanish at infinity.

657     We finally arrive at the result we were aiming to derive all along. The subalgebra $\mathfrak{G}_\infty$
658 generates (through the exponential map) the subgroup $\mathcal{G}_0^\infty \subset \mathcal{G}^I$ of gauge transformations
659 that become the identity at asymptotic infinity at the appropriate rate (which equals the rate
660 at which elements of $\mathcal{G}^I$ approach a constant) and lie in the identity component of $\mathcal{G}^I$ (i.e.
661 can be obtained by exponentiating Lie algebra elements). The quotient of physical gauge
662 transformations

$$\mathcal{G}_{\mathrm{DES}} = \mathcal{G}^I / \mathcal{G}_0^\infty$$

663 then looks like a copy of the global gauge group $G$, corresponding to all possible constants at
664 infinity, for every homotopy class of gauge transformations, i.e. for every connected compo-
665 nent of the gauge group $\mathcal{G}^I$. In three dimensions these homotopy classes are determined by the
666 fundamental group $\pi_3(G)$, since gauge transformations on $\Sigma$ that are constant at asymptotic
667 infinity can be viewed as maps $S^3 \to G$, where asymptotic infinity corresponds to one point on
668 $S^3$, e.g. the North pole. For $G = U(1)$ this homotopy group is trivial,[37] but for $G = SU(2)$ we
669 have $\pi_3(SU(2)) \cong \pi_3(S^3) \cong \mathbb{Z}$.

# 5   Adding the Higgs field

671 Over the past two decades there has been a substantial conceptual debate about the Higgs
672 mechanism [27, 68–72]. Much of this debate centers around the physical status of gauge
673 symmetries in relation to gauge symmetry breaking. As has been pointed out in [27, 30, 44,
674 73], a key point is that the unbroken and broken phases of the Higgs model exhibit differing
675 asymptotic boundary conditions. However, the derivation of this point has not been performed
676 rigorously. We can now do this in the framework developed in the previous Sections.
677     To include a Higgs field, we must enlarge the configuration space $Q_{A_\infty}$ of Yang-Mills fields
678 to $Q_{A_\infty} \times \tilde{Q}$, where $\tilde{Q}$ is the space of Higgs fields, which are sections of an associated vector
679 bundle $P \times_\rho V \to \Sigma$ through a representation $\rho \colon G \to \mathrm{GL}(V)$, where $V$ is the Higgs vector
680 space.[38] If we equip $V$ with an inner product $\langle \cdot, \cdot \rangle$, then we can define a norm on the sections
681 in $\Gamma(P \times_\rho V)$ in a similar way as for the gauge fields: by integrating the absolute value of such
682 a section over $\Sigma$.
683     The tangent space $T_\varphi \tilde{Q}$ at a point $\varphi \in \Gamma(P \times_\rho V)$ is itself just isomorphic to $\tilde{Q}$. However,
684 we need to restrict both $\tilde{Q}$ and $T\tilde{Q}$ with appropriate asymptotic boundary conditions. These
685 follow from the Yang-Mills-Higgs Lagrangian,[39] which is given by

$$\mathcal{L}_{\mathrm{YMH}}(A, \alpha_A, \varphi, \psi_\varphi) = \frac{1}{2} \|\alpha_A\|^2 - \frac{1}{2} \|F(A)\|^2 + \frac{1}{2} \|\psi_\varphi\|^2 - \frac{1}{2} \|D_A \varphi\|^2 - \int_\Sigma V(\varphi) d\mathrm{Vol},$$

686 for $A \in Q, \alpha_A \in T_A Q, \varphi \in \tilde{Q}, \psi_\varphi \in T_\varphi \tilde{Q}$. Here $V(\varphi)$ is the well-known Higgs potential, $D_A \varphi$ is
687 the covariant derivative of the Higgs field and $\psi_\varphi \in T_\varphi \tilde{Q}$ is thought of as the velocity of $\varphi \in \tilde{Q}$.
688     In order to guarantee finiteness of action and energy, we require that each individual term
689 in the above Lagrangian is finite. We already know that this requires $\alpha_A \to 0$ and $F(A) \to 0$,
690 but now we also need $\psi_\varphi \to 0$ and $D_A \varphi \to 0$, as well as a condition related to $V(\varphi)$. This last
691 condition is ambiguous. If the Higgs potential has the familiar shape

$$V(\varphi) = -\mu \|\varphi\|^2 + \lambda \|\varphi\|^4,$$

---

[37]In one dimension we do have interesting topology for electromagnetism since $\pi_1(S^1) \cong \mathbb{Z}$.

[38]It is $\mathbb{C}$ for $G = U(1)$, and $\mathbb{C}^2$ for both $G = SU(2)$ and $G = U(1) \times SU(2)$ [55].

[39]For the well-posedness of the Yang-Mills-Higgs initial value problem see [74].

then clearly the zero-point of $V(\varphi)$ lies at $\varphi = 0$ if $\mu > 0$ (besides some other manifold of roots at which $\varphi \neq 0$). Thus, we expect the boundary condition $\varphi \to 0$. However, we may instead want to think of the *minimum* of $V(\varphi)$ as the true vacuum, therefore requiring $\varphi \to$ min instead. These two possibilities respectively correspond to the so-called unbroken and broken phases of the Higgs model. Let us now study what the group $\mathcal{G}^I$ of boundary-preserving gauge symmetries looks like in both cases.[40]

*The unbroken phase.* In the unbroken phase, we assume that $\varphi = 0$ is the vacuum for the Higgs field, i.e. that this state carries zero energy. We can either think of this state as lying in the symmetric middle of the "Mexican hat potential," or as the potential itself being such that it only has a minimum at $\varphi = 0$, e.g. by taking $\mu < 0$. Since $\varphi = 0$ corresponds to zero energy, we require the asymptotic boundary condition $\varphi \to 0$, besides the common boundary conditions $\psi_\varphi \to 0$ and $D_A \varphi \to 0$ which are always needed. Note that this indeed gives the configuration space at infinity the right structure: the space of Higgs fields at infinity is zero-dimensional, since it consists only of $\varphi = 0$. The tangent space at infinity then also consists only of zero, which is what we want since we require $\psi_\varphi \to 0$. Now, the conditions $\psi_\varphi \to 0$ and $D_A \varphi \to 0$ are always preserved by any gauge transformation $g : \Sigma \to G$. This is obvious for the condition $D_A \varphi \to 0$, since the covariant derivative transforms covariantly via the linear Higgs represen­tation $\rho : G \to \mathrm{GL}(V)$. Similarly the condition $\psi_\varphi \to 0$ is preserved since $\psi_\varphi$ also transforms covariantly.[41] This means that the conditions $\psi_\varphi \to 0$ and $D_A \varphi \to 0$ are automatically pre­served, as 0 is mapped linearly to 0. The same goes for the condition $\varphi \to 0$, since $\varphi$ evidently transforms linearly under $\rho(g)$ with $g \in \mathcal{G}^I$. For pure Yang-Mills theory the group $\mathcal{G}^I$ consists of transformations that become constant asymptotically, and since the unbroken Higgs field boundary conditions do not impose any extra requirements on gauge transformations we find the same boundary-preserving gauge group for Yang-Mills-Higgs theory.

*The broken phase.* In the broken phase things are different. The asymptotic conditions are now $\psi_\varphi \to 0, D_A \varphi \to 0$ and $\varphi \to$ min. That is, the Higgs field must become a covariantly constant minimum of the potential $V(\varphi)$, and its velocity must become zero. At first sight, this seems to still allow $\mathcal{G}^I$ to contain all asymptotically constant transformations. After all, a gauge transformation maps a minimum of $V(\varphi)$ to another minimum. However, this is wrong, for the same reason as for pure Yang-Mills theory. Allowing for gauge transformations which act at infinity gives rise to a nontrivial configuration space $\tilde{Q}_\infty$ at infinity. After all, if we let $\varphi_\infty$ denote some covariantly constant minimum at infinity, then $\tilde{Q}_\infty$ will consist at least of an orbit of $\varphi_\infty$ under the action of $\mathcal{G}^I$ at infinity, i.e. of the constant gauge transformations at infinity. But this means that the tangent space $T_{\varphi_\infty} \tilde{Q}_\infty$ is far from being 0-dimensional. In fact, it has the dimension of $G$, since it is the tangent space to an orbit of the action of the constant gauge transformations, which orbit is isomorphic to $\mathfrak{g}$. But we cannot allow $T_{\varphi_\infty} \tilde{Q}_\infty$ to contain nonzero vectors, since we required that the tangent vectors $\psi_\varphi \in T_\varphi \tilde{Q}$ vanish at infinity! Like for pure Yang-Mills theory, the requirement that the Lagrangian be defined on the tangent bundle to configuration space forces us to put boundary conditions on the Higgs field itself, even though only the tangent vectors and curvatures appear in the Lagrangian.

Thus we are forced to require a stricter asymptotic boundary condition on the Higgs field: we need that $\varphi \to \varphi_\infty$, where $\varphi_\infty$ now denotes some *fixed*, covariantly constant minimum at infinity. This ensures that the configuration space at infinity is zero-dimensional, consisting only of $\varphi_\infty$. The tangent bundle at infinity is therefore also zero-dimensional, consisting only of $\psi_{\varphi_\infty} = 0$, as required for finiteness of energy. But clearly this stricter asymptotic boundary condition breaks gauge invariance, in the sense that it is not preserved by gauge

---

[40]We note that our ideas agree with [30], but fill in the missing argument, namely the Lagrangian must be defined on the tangent bundle to configuration space. Without this added argument one *cannot* deduce that the physical gauge group is different in the two cases.

[41]To see this, recall that, in covariant notation, we have $D_\mu \varphi \to \rho(g) \cdot D_\mu \varphi$, so in particular $D_0 \varphi \to \rho(g) \cdot D_0 \varphi$. In the 3+1 formalism we work in the temporal gauge $A_0 = 0$ and replace $D_0 \varphi = \partial_0 \varphi - e A_0 \varphi = \partial_0 \varphi$ by $\psi_\varphi$.

transformations which act non-trivially at infinity. Only gauge transformations that are the identity at infinity preserve $\varphi_\infty \neq 0$, so we find that the groups of boundary-preserving and redundant gauge symmetries are equal up to connected components, i.e. $\mathcal{G}^I = \mathcal{G}^\infty$, where $\mathcal{G}^\infty$ denotes the group of gauge transformations that are constant at infinity (but only its identity component $\mathcal{G}_0^\infty$ is generated by the Gauss law constraint).

In this way we conclude that the physical gauge group $\mathcal{G}^I/\mathcal{G}_0^\infty$ equals (several copies of) the global gauge group in the unbroken phase and is discrete (trivial in the Abelian case) in the broken phase. This conclusion results from the difference of what we call the "vacuum" in the two cases: either $\varphi = 0$ or a minimum of the potential $V(\varphi)$. Gauge symmetry breaking in the Higgs mechanism must therefore be understood as an alteration in the vacuum itself, leading to different asymptotic boundary conditions that ensure what it means to have finite energy.

## 6   Conclusion

In this paper we have given a rigorous derivation of the identification of (several copies of) the group of global gauge symmetries with the quotient of asymptotic symmetries

$$\mathcal{G}_{\mathrm{DES}} = \mathcal{G}^I/\mathcal{G}_0^\infty$$

in Yang-Mills theory on a three-dimensional Euclidean Cauchy surface. Here $\mathcal{G}^I$ denotes the group of allowed or boundary-preserving transformations and $\mathcal{G}_0^\infty$ the group of transformations that are trivial in the sense that they yield the Gauss law constraint through Noether's second theorem, and must therefore be viewed as redundant. Global gauge symmetries thus correspond to the asymptotic symmetry group with DES. There were two main points to this derivation, corresponding to obtaining $\mathcal{G}^I$ and $\mathcal{G}_0^\infty$ respectively.

Firstly, we found that instantaneous spatial asymptotic boundary conditions on Yang-Mills fields that ensure finiteness of energy only directly lead to requirements on tangent vectors $\alpha_A \in T_A Q$ and on the curvatures $F(A)$ of the connections $A \in Q$. However, as we want the domain of the Lagrangian to be the tangent bundle $TQ$ to the configuration space $Q$ of Yang-Mills fields, we need to also impose some asymptotic fall-off behavior on the gauge fields $A \in Q$ themselves. We have shown that it is not enough to require that gauge fields become flat at infinity, since this would still allow for non-zero tangent vectors $\alpha_A$ at infinity, which would spoil the finiteness of energy and action. Intuitively, this means gauge transformations acting at infinity create infinite energy, even if the energy depends only on gauge-invariant quantities. To counter this, we need to require that $A$ approaches a *fixed* flat connection at infinity. In the non-Abelian case we must also choose fixed flat connection to be invariant under the adjoint action of the structure group $G$. The gauge transformations that leave this fixed connection at infinity invariant are then precisely the elements of $\mathcal{G}$ that are constant at infinity. Properly interpreted, this means that we consider the equivalent problem on a conformal compactification $\hat{\Sigma}$ of $\Sigma$, and require that gauge transformations be constant on $\partial \hat{\Sigma}$. This yields the group of boundary-preserving gauge symmetries $\mathcal{G}^I$.

Secondly, we explained that redundant gauge transformations in the Hamiltonian formulation of Yang-Mills theory must be understood as the infinitesimal localizable symmetries $\mathfrak{G}$. These give rise to the Gauss law constraint and must therefore be interpreted as unphysical if (an appropriate form of) determinism is to survive. All infinitesimal symmetries in $\mathrm{Lie}(\mathcal{G}^I)$ are localizable, except for the global ones. Thus $\mathcal{G}_0^\infty$ indeed consists precisely of all gauge transformations that are the identity at infinity and lie in the identity component of $\mathcal{G}^I$. Again, properly interpreted this means we move to $\hat{\Sigma}$ from $\Sigma$ and require elements of $\mathfrak{G}$ to be zero on $\partial \hat{\Sigma}$, so that elements of $\mathcal{G}_0^\infty$ are the identity on $\partial \hat{\Sigma}$. The quotient $\mathcal{G}_{\mathrm{DES}}$ then consists of a

copy of the global gauge group $G$ for every homotopy class in $\pi_3(G)$.

Subsequently, we applied these ideas to Yang-Mills-Higgs theory, where we derived that $\mathcal{G}^I$ equals the group of asymptotically constant gauge transformations only in the unbroken phase. In the broken phase one can only permit asymptotically trivial transformations, for otherwise the action of the gauge group at infinity would create non-zero velocities of the Higgs field, carrying infinite energy.

In a subsequent article [75] we consider the implications of this last result for our interpretation of gauge symmetry breaking. There we argue that the Higgs mechanism must be understood as an instance of global gauge symmetry breaking, as has been proposed in the Abelian case for quantum electrodynamics [1,27,76–78]. In future research it would also be of interest to extend our results to spacetimes with a nonzero cosmological constant and to better understand the relation of our work to asymptotic symmetries of Yang-Mills fields on the full boundary of spacetime (e.g. in celestial holography) [10], as well as to edge modes [8,79,80] and (quantum) reference frames [81–83], and boundaries which are not asymptotic [4–7,9]. Additionally, it may be conceptually cleaner to reformulate our derivation in the language of jet bundles. Finally, the implications for quantum field theory should be investigated [84].

# Acknowledgements

The authors want to sincerely thank Klaas Landsman and Manus Visser for providing detailed feedback on manuscripts, suggesting technical and conceptual improvements and for their general involvement and guidance. They are also grateful to Sebastian De Haro for his collaboration on this and related work. SB thanks Aldo Riello and Henrique Gomes for their comments and the insightful conversations in real life and over email.

**Author contributions**   SB wrote the article, HB contributed with discussions, feedback and supervision.

**Funding information**   This work is supported by the Spinoza Grant of the Dutch Science Organization (NWO) awarded to N.P. (Klaas) Landsman.

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
