# Peer review of "Global Gauge Symmetries and Spatial Asymptotic Boundary Conditions in Yang-Mills Theory"

_SciPost Physics_

## Round 1 · Referee Report · Anonymous (Referee 1) · 2025-10-11

Strengths

  1. The manuscript convincingly argues that it addresses a significant and long-standing open problem in the precise understanding of asymptotic symmetries in Yang-Mills theory.

  2. It brings in relevant and powerful mathematical technology, such as the formalism of principal bundles and the symplectic geometry of the phase space, to tackle the problem.

Weaknesses

  1. The central argument, which is primarily mathematical, is presented in a way that is difficult to follow and verify. The logic is spread across lengthy prose rather than being clearly structured.

  2. The presentation appears to be affected by several fundamental misconceptions (e.g., regarding the concepts of gauge, Lagrangian, and logical inference from isomorphisms), which casts doubt on the reliability of the conclusions.

Report

This article aims to resolve subtle issues in the formulation of asymptotic falloff conditions on gauge fields, which are crucial for a precise definition of asymptotic symmetries in Yang-Mills theories. Given the foundational importance of both Yang-Mills theory and the study of asymptotic symmetries, this is a highly valuable and timely goal. Addressing potential gaps in the community's understanding of this topic is a commendable effort.

The subject matter is squarely in the domain of mathematical physics, where precision and logical clarity are paramount. The main claims of the paper rely on detailed mathematical arguments where rigor is essential to validate the physics intuition.

In light of this, it is unfortunate that the manuscript's presentation does not meet the expected standards of clarity for a work of this nature. The core arguments are developed over many pages of continuous prose, making it exceedingly difficult for a reader to trace, validate, and build confidence in the logical flow. The paper would be substantially improved by adopting a more transparent structure, such as a Definition/Proposition/Proof format. This would allow the authors to state their precise claims in clearly marked propositions and subsequently provide the detailed arguments in distinct, self-contained proofs. Such a structure serves to guide the reader and makes the authors' novel contributions stand out clearly from standard textbook material.

This structural issue is compounded by several instances of imprecise terminology and flawed reasoning, which undermine the reader's confidence in the overall analysis. Notable examples include:

  • Conflating the notion of a "gauge choice" with a "trivialization of the principal bundle" (a global section).

  • An unconventional use of the term "Lagrangian" for a spatially integrated density, and "action" for its time integral.

  • A recurring logical error where properties are claimed to be transferred between spaces based merely on the existence of a vector space isomorphism between them.

For these reasons, I cannot recommend the article for publication in its present form. However, the topic is important, and it is possible that the authors' underlying physical and mathematical results are sound. If so, a major revision that focuses on communicating the arguments clearly and rigorously would be of great benefit to the community. Such a rewrite would not only serve the reader but also strengthen the authors' own case.

Validity: Unclear. The central claims depend on a mathematical argument that is not presented with sufficient clarity or rigor to be verified.

Significance: The paper addresses a foundational question in a very important area of theoretical physics.

Originality: The approach uses known techniques, but the specific application to clarify these foundational issues appears to be novel.

Clarity: This is the primary weakness of the manuscript, as detailed in the report.

Requested changes

A substantial rewrite is necessary to address the structural and conceptual issues detailed in the report. The primary goal should be to present the mathematical arguments in a clear, structured, and verifiable manner. Below is a list of specific points that must be addressed.

General:

Structure the Argument: Reformat the core mathematical claims and their derivations into a clear Proposition/Proof structure. This should be applied to the novel results of the paper, not to standard textbook facts (which should be cited instead).

Rectify Terminology: Carefully review and correct the use of fundamental terms, including "gauge", "trivialization", "Lagrangian", and "action", to align with standard usage in mathematical physics.

Specific Points from the Text:

p. 6: The text states, "We do this without working in a particular gauge," but then immediately declares, "we work in the temporal gauge." This apparent contradiction should be resolved, and the intended meaning clarified.

p. 6, line 203: The definition "a gauge, i.e., a section s --> P" refers to a global section, which is a trivialization of the bundle. This is a very strong assumption. If a global trivialization is indeed assumed throughout, the formalism involving the adjoint bundle is largely unnecessary and should be simplified. If it is not assumed, the terminology and arguments must be corrected to properly handle potentially non-trivial bundles (e.g., by working with local sections). This confusion reappears in line 454.

p. 7, line 222: Proposition 2.1 and Theorem 2.2 are standard textbook results. Presenting them as the only formally numbered theorems in the paper is misleading. These should be cited, and the authors' own key results should be presented as propositions or theorems.

p. 7, line 247: The use of "Lagrangian" to refer to a quantity that is already integrated over all of space is non-standard. In a paper striving for mathematical precision, the distinction between a Lagrangian density (the integrand), a Lagrangian (the spatial integral, in Hamiltonian formalism), and the action (the spacetime integral) should be made clear and used consistently.

p. 11, lines 379-380 and 416-417: The argument that asymptotic properties are transferred between spaces "by the isomorphism" (e.g., TAQ≅Q) is not valid. The mere fact that a pair of spaces (affine space, in the present case) is isomorphic does not entail any statement about general pairs of elements in them. The reasoning for why the falloff conditions for tangent vectors imply similar conditions for the connections themselves needs to be stated carefully and correctly.

p. 13, line 455: The statement "Let us, for simplicity, assume P has now been trivialized..." is problematic. It gives the impression that the preceding bundle-theoretic discussion was purely cosmetic. The authors must either (a) assume from the start that the bundle is trivial and remove the unnecessary general formalism, or (b) carry out their analysis for the general case of a possibly non-trivial bundle.

pp. 13-15: This lengthy review of the symplectic formulation of Yang-Mills theory should either be streamlined and cited from a standard reference (e.g., [48] in the authors' own bibliography) or explicitly framed as a pedagogical review necessary for the subsequent steps.

p. 22, line 805: There appears to be a typo in the second author's initials in the "Author contributions" section (should likely be HBP, not HB).

Recommendation

Ask for major revision

---

## Round 1 · Referee Report · Anonymous (Referee 2) · 2025-10-20

Disclosure of Generative AI use

The referee discloses that the following generative AI tools have been used in the preparation of this report:

I submitted my comments to ChatGpt5 and asked it to formulate in a format adequate for Scipost specifications.

Strengths

  1. The paper addresses a timely and conceptually important topic: the derivation of the physical gauge group in Yang–Mills theory from first principles, particularly in relation to finite-energy and asymptotic boundary conditions.
  2. The presentation is clear, well-structured, and pedagogically useful; it reviews the relevant geometric and analytic background in a form accessible to readers outside the subfield.
  3. The discussion of the conformal compactification technique and its use in regularizing asymptotic conditions is careful and technically competent.
  4. The attempt to link the restriction to boundary-preserving gauge transformations to the mathematical structure of the Lagrangian—rather than postulating it—is original and philosophically well-motivated.
  5. The treatment of the Yang–Mills–Higgs case, distinguishing between broken and unbroken phases, broadens the relevance of the results beyond the Abelian prototype.

Weaknesses

  1. The central claim—that one must insist on a cotangent-bundle structure (i.e. (T^*Q)) to justify the restriction to boundary-preserving gauge transformations—is asserted rather than demonstrated. No argument is given that the finite-energy constraint cannot simply be treated as defining a submanifold of the large phase space.
  2. The mathematical reasoning is incomplete: the analysis never checks whether the finite-energy constraint surface is coisotropic with respect to the symplectic form.
  3. The interplay between the Gauss constraint and the finite-energy condition is not analysed geometrically. These two constraints should be treated jointly, since only their intersection defines the physically relevant phase space.
  4. The argument linking fall-off of the electric field to fall-off of the potential (A_i) conflates coordinate and gauge-dependent descriptions. The transition from (F_{ij}) (gauge-invariant) to (A_i) (gauge-variant) requires either a fixed gauge or an explicit discussion of gauge equivalence classes. If energy is entirely specified by E and B (or F), why couldn't fall off conditions be described purely in terms of E and B (or F)?
  5. The supposed restriction on the gauge group therefore remains gauge-dependent and conceptually ambiguous. The statement that certain gauge transformations are “not allowed” is not substantiated in the presymplectic framework.
  6. The treatment of the asymptotic gauge parameters and boundary terms in the momentum-map calculation is heuristic: different fall-off assumptions yield different “allowed” classes of transformations. The resulting quotient group (G_I/G_\infty^0) is not unambiguously fixed.

Report

The manuscript makes a valuable effort to clarify a subtle point in gauge theory: how the physical gauge group arises once finite-energy conditions and Lagrangian structure are imposed. However, the main argument currently rests on an unproven assumption—that one must maintain a cotangent-bundle structure after restriction. A more mathematically and conceptually coherent strategy would proceed via presymplectic geometry.

Let ( (P, \omega) ) be the large (infinite-dimensional) symplectic manifold of fields and their conjugate momenta. The finite-energy condition and Gauss’s law together define a constraint surface ( C \subset P ). The correct question is whether (C) is *coisotropic*, i.e. whether
[
(TC)^{\perp_\omega} \subset TC .
]
If this holds, one can perform coisotropic (Dirac) reduction to obtain the reduced phase space ( C/{!\sim} ) endowed with a natural symplectic structure. In this framework, all gauge transformations generated by the Gauss constraint are automatically tangent to (C) and lie in the kernel of the restricted form; they therefore represent redundant directions. Only boundary (non-vanishing) gauge transformations need special treatment.

Adopting this geometric perspective would eliminate the need to stipulate, *a priori*, a specific cotangent-bundle structure or to exclude gauge transformations “by hand.” It would also clarify the status of asymptotic conditions and boundary flux terms, which can then be handled systematically within the presymplectic formalism.

Overall, the paper’s aims are worthwhile, but its main derivation should be reformulated within this standard geometric framework.

Requested changes

  1. Reformulate the main argument using the language of presymplectic geometry. Explicitly define the constraint surface (C) given by finite-energy and Gauss constraints, and check whether (C) is coisotropic.
  2. Replace the assumption that (TQ) must underlie a cotangent-bundle structure with a derivation based on restricting the symplectic form to the constraint surface.
  3. Explain how the finite-energy and Gauss constraints interact and what happens if their intersection fails to be coisotropic.
  4. Distinguish clearly between gauge-invariant (field-strength-based) and gauge-dependent (potential-based) fall-offs. Explain how any restriction on (A_i) can be stated in gauge-invariant form or under explicit gauge fixing.
  5. Show explicitly how gauge transformations that vanish at the boundary correspond to null directions of the presymplectic form, and why no further restriction is necessary.
  6. Give a consistent account of how the electric flux term behaves under conformal compactification and what its vanishing requires of gauge parameters.
  7. If the finite-energy constraint turns out not to be coisotropic, describe what physical or mathematical consequences follow.

Recommendation

Ask for major revision

---

## Editorial Decision

awaiting_resubmission